# Localizing Memorization in SSL Vision Encoders

**Wenhao Wang[1], Adam Dziedzic[1], Michael Backes[1], Franziska Boenisch[1]***
[1]CISPA, Helmholtz Center for Information Security

## Abstract

Recent work on studying memorization in self-supervised learning (SSL) suggests that even though SSL encoders are trained on millions of images, they still memorize individual data points. While effort has been put into characterizing the memorized data and linking encoder memorization to downstream utility, little is known about where the memorization happens inside SSL encoders. To close this gap, we propose two metrics for localizing memorization in SSL encoders on a per-layer (`LayerMem`) and per-unit basis (`UnitMem`). Our localization methods are independent of the downstream task, do not require any label information, and can be performed in a forward pass. By localizing memorization in various encoder architectures (convolutional and transformer-based) trained on diverse datasets with contrastive and non-contrastive SSL frameworks, we find that (1) while SSL memorization increases with layer depth, highly memorizing units are distributed across the entire encoder, (2) a significant fraction of units in SSL encoders experiences surprisingly high memorization of individual data points, which is in contrast to models trained under supervision, (3) *atypical* (or outlier) data points cause much higher layer and unit memorization than standard data points, and (4) in vision transformers, most memorization happens in the fully-connected layers. Finally, we show that localizing memorization in SSL has the potential to improve fine-tuning and to inform pruning strategies.

## 1 Introduction

Self-supervised learning (SSL) ([16, 17, 13, 4, 30, 27, 29]) enables pre-training large encoders on unlabeled data to generate feature representations for a multitude of downstream tasks. Recently, it was found that, even though their training datasets are large, SSL encoders still memorize individual data points ([36, 47]). While prior work characterizes the memorized data and studies the effect of memorization to improve downstream generalization ([47]), little is known about where in SSL encoders memorization happens.

The few works on localizing memorization are usually confined to supervised learning (SL) ([3, 45, 35]), or operate in the language domain ([55, 37, 7, 44, 14]). In particular, most results are coarse-grained and localize memorization on a per-layer basis [3, 45] and/or require labels [3, 35].

To close the gap, we propose two novel metrics for localizing memorization in SSL encoders in the vision domain. Our `LayerMem` localizes memorization of the training data within the SSL encoders on a layer-level. For a more fine-grained localization, we turn to memorization in individual *units* (*i.e.,* neurons in fully-connected layers or channels in convolutional layers). We propose `UnitMem` which measures memorization of individual training data points through the units' sensitivity to these points. Both our metrics can be computed independently of a downstream task, in a forward pass without gradient calculation, and without labels, which makes them computationally efficient and well-suited for the large SSL encoders pretrained on unlabeled data. By performing a systematic study on localizing memorization with our two metrics on various encoder architectures (convolutional

---

*Correspondence to boenisch@cispa.de

*Accepted for publication at:* 38th Conference on Neural Information Processing Systems (NeurIPS 2024).

and transformer-based) trained on diverse vision datasets with contrastive and non-contrastive SSL frameworks, we make the following key discoveries:

**Memorization happens through the entire SSL encoder.** By analyzing our `LayerMem` scores between subsequent layers, we find that the highest memorizing layers in SSL are not necessarily the last ones, which is in line with findings recently reported for SL [35]. While there is a tendency that higher per-layer memorization can be observed in deeper layers, similar to SL [3, 45], our analysis of memorization on a per-unit level highlights that highly memorizing units are distributed across the entire SSL encoder, and can also be found in the first layers.

**Units in SSL encoders experience high memorization.** By analyzing SSL encoders with our `UnitMem` metric, we find that a significant fraction of their units experiences high memorization of individual training data points. This stands in contrast with models trained using SL for which we observe high class memorization, measured as the unit's sensitivity to any particular class. While these results are in line with the two learning paradigms' objectives where SL optimizes to separate different classes whereas SSL optimizes foremost for instance discrimination [48], it is a novel discovery that this yields significantly different memorization patterns between SL and SSL down to the level of individual units.

**Atypical data points cause higher memorization in layers and units.** While prior work has shown that SSL encoders overall memorize atypical data points more than standard data points [47], our study reveals that the effect is constant throughout *all* encoder layers. Hence, there are no particular layers responsible for memorizing atypical data points, similarly as observed in SL [35]. Yet, memorization of atypical data points can be attributed on a unit-level where we observe that the highest memorizing units align with the highest memorized (atypical) data points and that overall atypical data points cause higher unit memorization than standard data points.

**Memorization in vision transformers happens mainly in the fully-connected layers.** The memorization of transformers [46] was primarily investigated in the language domain [26, 43]. However, the understanding in the vision domain is lacking, and due to the difference in input and output tokens (language transformers operate on discrete tokens while vision transformers operate on continuous ones), the methods for analysis and the findings are not easily transferable. Yet, with our methods to localize memorization, we are the first to show that the same trend holds in vision transformers that was previously reported for language transformers, namely that memorization happens in the fully-connected layers.

Finally, we investigate future applications that could benefit from localizing memorization and identify *more efficient fine-tuning* and *memorization-informed pruning strategies* as promising directions.

In summary, we make the following contributions:

- We propose `LayerMem` and `UnitMem`, the first practical metrics to localize memorization in SSL encoders on a per-layer basis and down to the granularity of individual units.
- We perform an extensive experimental evaluation to localize memorization in various encoder architectures trained on diverse vision datasets with different SSL frameworks.
- Through our metrics, we gain new insights into the memorization patterns of SSL encoders and can compare them to the ones of SL models.
- We show that the localization of memorization can yield practical benefits for encoder fine-tuning and pruning.

## 2   Related Work

**SSL.** SSL relies on large amounts of unlabeled data to train encoder models that return representations for a multitude of downstream tasks [6]. Especially in the vision domain, a wide range of SSL frameworks have recently been introduced [16, 17, 13, 4, 30, 27, 29]. Some of them rely on contrastive loss functions [16, 29, 27] whereas others train with non-contrastive objective functions [41, 17, 13, 30].

**Memorization in SL.** Memorization was extensively studied in SL [52, 2, 15]. In particular, it was shown that it can have a detrimental effect on data privacy, since it enables data extraction attacks [9, 10, 11]. At the same time, memorization seems to be required for generalization, in

particular for long-tailed data distributions [23, 24]. It was also shown that *harder* or more atypical data points [2, 43] experience higher memorization. While all these works focus on studying memorization from the data perspective and concerning its impact on the learning algorithm, they do not consider where memorization happens.

**Memorization in SSL.** Even though SSL rapidly grew in popularity during recent years, work on studying memorization in SSL is limited. Meehan et al. [36] proposed to quantify Déjà Vu memorization of SSL encoders with respect to particular data points by comparing the representations of these data points with the representations of a labeled public dataset. Data points whose $k$ nearest public neighbors in the representation space are highly consistent in labels are considered to be memorized. Since SSL is aimed to train *without labels*, this approach is limited in practical applicability. More recently, Wang et al. [47] proposed `SSLMem`, a definition of memorization for SSL based on the leave-one-out definition from SL [23, 24]. Instead of relying on labels, this definition captures memorization through representation alignment, *i.e.,* measuring the distance between representations of a data point's multiple augmentations. Since both works rely on the output representations to quantify memorization, neither of them is suitable for performing fine-grained localization of memorization. Yet, we use the setup of `SSLMem` as a building block to design our `LayerMem` metric which localizes memorization per layer.

**Localizing Memorization.** In SL, most work focuses on localizing memorization on a per-layer basis and suggests that memorization happens in the deeper layers [3, 45]. By analyzing which neurons have the biggest impact on *predicting the correct label* of a data point, Maini et al. [35] were able to study memorization on a per-unit granularity. They do so by zero-ing out random units until a label flip occurs. Their findings suggest that only a few units are responsible for memorizing outlier data points. Yet, due to the absence of labels in SSL, this approach is inapplicable to our work. In the language domain, a significant line of work aims at localizing where semantic facts are stored within large language transformers [55, 37, 7, 44]. Chang et al. [14] even proposed *benchmarks* for localization methods in the language domain. In the injection benchmark (INJ Benchmark), they fine-tune a small number of neurons and then assess whether the localization method detects the memorization in these neurons. The deletion benchmark (DEL Benchmark) first performs localization, followed by the deletion of the responsible neurons, and a final assessment of the performance drop on the data points detected as memorized in the identified neurons. Since in SSL, performance drop cannot be measured directly due to the absence of a downstream task, the deletion approach is not applicable. Instead, we verify our `UnitMem` metric in a similar vein to the INJ Benchmark by fine-tuning a single unit on a data point and localizing memorization as we describe in Section 5.2.

**Studying Individual Units in ML Models.** Early work in SL [22] already suggested that units at different model layers fulfill different functions: while units in lower layers are responsible for extracting general features, units in higher layers towards the model output are responsible for very specific features [51]. In particular, it was found that units represent different concepts required for the primary task [5], where some units focus on single concepts whilst others are responsible for multiple concepts [38, 54]. While these differences have been identified between the units of models trained with SL, we perform a corresponding investigation in the SSL domain through the lens of localizing memorization.

## 3 Background and Setup

**SSL and Notation.** We consider an SSL training framework $\mathcal{M}$. The encoder $f : \mathbb{R}^n \to \mathbb{R}^s$ is pre-trained, or in short *trained*, on the unlabeled dataset $\mathcal{D}$ to output representations of dimensionality $s$. Throughout the training, as the encoder improves, its alignment loss $\mathcal{L}_{\mathcal{A}}(f, x) = d(f(x'), f(x''))$ between the representations of two random augmentations $x', x''$ of any training data point $x$ decreases with respect to a distance metric $d$ (*e.g.,* Euclidean distance). This effect has also been observed in non-contrastive SSL frameworks [53]. We denote by $f^l, l \in [1, \ldots L]$ the $l^{th}$ layer of encoder $f$. Data points from the test set $\bar{\mathcal{D}}$ are denoted as $\bar{x}$.

**Memorized Data.** Prior work in the SL domain usually generates outliers for measuring memorization by flipping the labels of training data points [23, 24, 35]. This turns these points into outliers

that experience a higher level of memorization and leave the strongest possible signal in the model. Yet, such an approach is not suitable in SSL where labels are unavailable. Therefore, we rely on the `SSLMem` metric proposed by [47] to identify the most (least) memorized data points for a given encoder. The findings based on the `SSLMem` metric indicate that the most memorized data points correspond to atypical and outlier samples.

**`SSLMem` for Quantifying Memorization.** `SSLMem` quantifies the memorization of individual data points by SSL encoders. It is, to the best of our knowledge, the only existing method for quantifying memorization in SSL without reliance on downstream labels. `SSLMem` for a training data point $x$ is defined as

$$\text{SSL}_f(x) = \underset{f\sim\mathcal{M}(\mathcal{D})}{\mathbb{E}} \underset{x',x''\sim\text{Aug}(x)}{\mathbb{E}} d\left(f(x'), f(x'')\right); \text{SSL}_g(x) = \underset{g\sim\mathcal{M}(\mathcal{D}\backslash x)}{\mathbb{E}} \underset{x',x''\sim\text{Aug}(x)}{\mathbb{E}} d\left(g(x'), g(x'')\right)$$
$$\text{SSLMem}_{f,g}(x) = \text{SSL}_g(x) - \text{SSL}_f(x) \tag{1}$$

where $f$ and $g$ are two classes of SSL encoders whose training dataset $\mathcal{D}$ differs in data point $x$. $x'$ and $x''$ denote two augmentations randomly drawn from the augmentation set *Aug* that is used during training and $d$ is a distance metric, here $\ell_2$-distance.

**Experimental Setup.** We localize memorization in encoders trained with different SSL frameworks on five common vision datasets, namely CIFAR10, CIFAR100, SVHN, STL10, and ImageNet. We leverage different model architectures from the ResNet family, including ResNet9, ResNet18, ResNet34, and ResNet50. We also analyze Vision Transformers (ViTs) using their Tiny and Base versions. Results are reported over three independent trials. To identify the most memorized training data points, we rely on the `SSLMem` metric and follow the setup from [47]. More details on the experimental setup can be found in Appendix B.[2] For the readers' convenience, we include a glossary with short explanations for all concepts and background relevant to this work in Appendix A.

## 4   Layer-Level Localization of Memorization

In order to localize memorization on a per-layer granularity, we propose a new `LayerMem` metric which relies on the `SSLMem` metric, as a building block. Since the `SSLMem` as defined in Equation (1) is not normalized, we introduce the following normalization to the range $[0, 1]$

$$\text{SSLMem}'_{f,g}(x) = \frac{\text{SSL}_g(x) - \text{SSL}_f(x)}{\text{SSL}_f(x) + \text{SSL}_g(x)} \tag{2}$$

such that values close to $0$ denote no memorization while $1$ denotes the highest memorization. This makes the score more interpretable. While `SSLMem` returns a memorization score per data point for a given encoder, `LayerMem` returns a memorization score per encoder layer $l$, measured on a (sub)set $\mathcal{D}' = \{x_1, ..., x_{|\mathcal{D}'|}\} \subseteq \mathcal{D}$ of training data $\mathcal{D}$. Similar to `SSLMem`, `LayerMem` makes use of a second encoder $g$ as a reference to detect memorization as

$$\text{LayerMem}_{\mathcal{D}'}(l) = \frac{1}{|\mathcal{D}'|} \sum_{i=1}^{|\mathcal{D}'|} \text{SSLMem}'_{f^l, g^l}(x_i). \tag{3}$$

$f^l, g^l$ denote the output of encoders $f$ and $g$ after layer $l$, respectively. Intuitively, our `LayerMem` metric measures the average per-layer memorization over training data points $x_i \in \mathcal{D}'$. As our `LayerMem` build on `SSLMem` ', it also inherits the above normalization. Since Equation (3) operates on different layers' outputs which in turn depend on all previous layers, `LayerMem` risks to report accumulated memorization up to layer $l$. Therefore, we also define $\Delta\,\text{LayerMem}_{\mathcal{D}'}(l)$ for all layers $l > 1$ as

$$\Delta\text{LayerMem}_{\mathcal{D}'}(l) = \text{LayerMem}_{\mathcal{D}'}(l) - \text{LayerMem}_{\mathcal{D}'}(l - 1). \tag{4}$$

This reports the increase in memorization of layer $l$ with respect to the previous layer $l - 1$.

---

[2]Our code is attached as supplementary material.

### 4.1 Experimental Results and Observations

We present our core results and provide additional ablations on our `LayerMem` in Appendix C.3.

**Memorization Increases but not Monotonically.** We report the `LayerMem` scores in Table 1 for the ResNet9-based SSL encoder trained with SimCLR on CIFAR10 (further per-layer breakdown and scores for ResNet18, ResNet34, and ResNet50 are presented in Table 15, Table 16, and Table 17 in Appendix C.3). We report `LayerMem` across the 100 randomly chosen training data points, their $\Delta$`LayerMem` (denoted as $\Delta$LM), followed by `LayerMem` for only the Top 50 memorized data points, their $\Delta$`LayerMem` (denoted as $\Delta$LM Top50), and `LayerMem`

Table 1: **Layer-based Memorization Scores.** Res$N$ denotes a residual connection that comes from the previous $N$-th convolutional layer.

| Layer | LayerMem | $\Delta$LM | LayerMem Top50 | $\Delta$LM Top50 | LayerMem Least50 |
|---|---|---|---|---|---|
| 1 | 0.091 | - | 0.144 | - | 0.003 |
| 2 | 0.123 | 0.032 | 0.225 | 0.081 | 0.012 |
| 3 | 0.154 | 0.031 | 0.308 | 0.083 | 0.022 |
| 4 | 0.183 | 0.029 | 0.402 | 0.094 | 0.031 |
| Res2 | 0.185 | 0.002 | 0.403 | 0.001 | 0.041 |
| 5 | 0.212 | 0.027 | 0.479 | 0.076 | 0.051 |
| 6 | 0.246 | 0.034 | 0.599 | 0.120 | 0.061 |
| 7 | 0.276 | 0.030 | 0.697 | 0.098 | 0.071 |
| 8 | 0.308 | 0.032 | 0.817 | 0.120 | 0.073 |
| Res6 | 0.311 | 0.003 | 0.817 | 0 | 0.086 |

for only the Least 50 memorized data points. The results show that our `LayerMem` indeed increases with layer depth in SSL, similar to the trend observed for SL [45], *i.e.,* deeper layers experience higher memorization than early layers. However, our $\Delta$`LayerMem` presents the memorization from a more accurate perspective, where we discard the accumulated memorization from previous layers, including the residual connections. $\Delta$`LayerMem` indicates that the memorization increases in all the layers but is not monotonic.

We also study the differences in localization of the memorization for most memorized (outliers and atypical examples) vs. least memorized data points (inliers), shown as columns `LayerMem` Top50 and `LayerMem` Least50 in Table 1, respectively. While we observe that the absolute memorization for the most memorized data points is significantly higher than for the least memorized data points, they both follow the same trend of increasing memorization in deeper layers. The $\Delta$`LayerMem` for the most memorized points (denoted as $\Delta$LM Top50 in Table 1) indicates that, following the overall trend, high memorization of the atypical samples is also spread over the entire encoder and not confined to particular layers.

**Memorization in Vision Transformers.** The memorization of Transformers [46] was, so far, primarily investigated in the language domain [26, 43], however, its understanding in the vision domain is lacking. The fully-connected layers in *language transformers* were shown to act as key-value *memories*. Still, findings from language transformers cannot be easily transferred to *vision transformers* (ViTs) [20]: while language transformers operate on the level of *discrete* and interpretable input and output tokens, ViTs operate on *continuous* input image patches and output representations. Through the analysis of our newly proposed metric for memorization in SSL, in Table 2 (ViT-Tiny trained on CIFAR10 using MAE [30]), we are the

Table 2: **Memorization in ViT occurs primarily in the deeper blocks and more in the fully connected than attention layers.**

| ViT Block | LayerMem | $\Delta$LayerMem |
|---|---|---|
| *Attention Layer* | | |
| 2 | 0.028 | 0.008 |
| 6 | 0.114 | 0.009 |
| 12 | 0.281 | 0.010 |
| *Fully-Connected Layer* | | |
| 2 | 0.039 | 0.011 |
| 6 | 0.129 | 0.015 |
| 12 | 0.303 | 0.022 |

first to show that memorization in ViTs occurs more in deeper blocks and that within the blocks, *fully-connected layers memorize more than attention layers*. We present the full set of results for `LayerMem` and $\Delta$`LayerMem` over *all blocks* in Table 10.

**Memorization in Different SSL Frameworks.** We also study the differences in memorization behavior between different SSL frameworks. Therefore, we compare the `LayerMem` score between corresponding layers of a ResNet50 trained on ImageNet with SimCLR [16] and DINO [13], and of a ViT-Base encoder trained on ImageNet with DINO and MAE [30]. We ensure by early stopping that the resulting linear probing accuracies of the encoder pairs are similar for better comparability of their memorization. The ImageNet downstream task performance within both encoder pairs is 66.12% for SimCLR and 68.44% for DINO; and 60.43% for MAE and 60.17% for DINO. Our results in Table 4 show that encoders with the same architecture trained with different SSL frameworks experience a similar memorization pattern, namely that memorization occurs primarily in the deeper blocks/layers. In Figure 12 in Appendix C.1, we additionally show that memorization patterns between different

Table 3: **Consistency in 100 most memorized samples according to** `LayerMem`**.** We report the pairwise overlap between the most memorized samples and the consistency in ranking of most memorized samples using the statistical Kendall's Tau test ($\tau$-statistic, $p$-value). While we observe high overlap and statistical similarity within adjacent layers, especially towards the end of the network, there is low similarity and overlap between early and late layers.

| Layers | Overlap % | $\tau, p$ | Layers | Overlap % | $\tau, p$ | Layers | Overlap % | $\tau, p$ | Layers | Overlap % | $\tau, p$ |
|---|---|---|---|---|---|---|---|---|---|---|---|
| **1 2** | 79 | 0.607, 4.07e-29 | **1 3** | 52 | 0.505, 9.08e-11 | **1 4** | 47 | 0.412, 6.07e-7 | **1 5** | 24 | 0.240, 1.18e-2 |
| **1 6** | 19 | 0.181, 6.01e-2 | **1 7** | 18 | 0.167, 744e-2 | **1 8** | 16 | 0.104, 1.27e-1 | **2 3** | 70 | 0.562, 8.48e-19 |
| **2 4** | 64 | 0.544, 3.95e-16 | **2 5** | 36 | 0.288, 2.08e-4 | **2 6** | 30 | 0.249, 3.96e-3 | **2 7** | 28 | 0.241, 9.96e-3 |
| **2 8** | 27 | 0.247, 5.53e-3 | **3 4** | 82 | 0.665, 4.41e-42 | **3 5** | 51 | 0.512, 6.67e-11 | **3 6** | 42 | 0.356, 8.31e-5 |
| **3 7** | 39 | 0.319, 8.31e-5 | **3 8** | 37 | 0.310, 1.09e-4 | **4 5** | 68 | 0.557, 1.61e-18 | **4 6** | 54 | 0.509, 6.31e-11 |
| **4 7** | 48 | 0.412, 6.67e-7 | **4 8** | 45 | 0.396, 2.50e-6 | **5 6** | 72 | 0.559, 4.11e-18 | **5 7** | 61 | 0.531, 4.19e-14 |
| **5 8** | 58 | 0.527, 1.08e-14 | **6 7** | 84 | 0.657, 1.47e-42 | **6 8** | 79 | 0.644, 4.17e-37 | **7 8** | 94 | 0.837, 9.71e-76 |

SSL frameworks are similar down to the individual unit level, *i.e.,* the number of highly memorizing units and the magnitude of memorization are roughly the same. We present the full results for *all* ResNet50 layers and *all* ViT blocks in Table 29 and Table 31, respectively.

**Variability and Consistency of Memorization cross Different Layers.** We use `LayerMem` to analyze the variability and consistency between the samples memorized by different layers in a ResNet9 vision encoder trained with CIFAR10 dataset. The results are shown in Table 3 and Figure 13 in appendix C.5. The overlap within the 100 most memorized samples between adjacent layers is usually high but decreases the further the layers are separated. Our statistical analysis to compare the similarity of the orderings within different layers' most memorized samples using the Kendall's rank correlation coefficient shows that while for closer layers, we manage to reject the null hypothesis ("no correlation") with high statistical confidence (low p-value) which is not the case for further away layers.

## 4.2 Verification of Layer-Based Memorization

To analyze whether our `LayerMem` metric and its $\Delta$ variant indeed localize memorization correctly, we first replace different layers of an encoder and then compute linear probing accuracy on various downstream tasks. Since prior work shows that memorization in SSL is required for downstream generalization [47], we expect the highest performance drop when replacing the layers identified as most memorizing.

Our results in Appendix C.7 verify this intuition. They show that by replacing *the most memorizing layers* of an encoder trained on

Table 4: **The layer-based memorization is similar across encoders trained with different frameworks.** LM=`LayerMem`, $\Delta$LM=$\Delta$`LayerMem`.

| ResNet50 Layer | *SimCLR* | | *DINO* | |
| Number | LM | $\Delta$LM | LM | $\Delta$LM |
|---|---|---|---|---|
| 2 | 0.040 | 0.003 | 0.041 | 0.002 |
| 27 | 0.161 | 0.005 | 0.165 | 0.006 |
| 49 | 0.302 | 0.008 | 0.311 | 0.007 |

| ViT-Base Block | *MAE* | | *DINO* | |
| Number | LM | $\Delta$LM | LM | $\Delta$LM |
|---|---|---|---|---|
| 2 | 0.037 | 0.010 | 0.036 | 0.011 |
| 6 | 0.120 | 0.015 | 0.116 | 0.014 |
| 12 | 0.274 | 0.019 | 0.271 | 0.019 |

a dataset $A$, *e.g.,* CIFAR10, with the equivalent layers of another dataset $B$, *e.g.,* STL10, the linear probing accuracy drop for CIFAR10 is significantly larger than when when *replacing random or least memorizing layers*. Surprisingly, at the same time, the replacement of the most memorizing layers from the CIFAR10 trained encoder with STL10 layers also causes the highest increase in STL10 linear probing accuracy (again in comparison to replacing random or least memorizing layers). See a full set of results for replacing any combination of 1, 2, and 3 layers in Table 30, Table 32, and Table 33, respectively. These results suggest that we might be able to improve standard encoder fine-tuning by localizing the most memorizing layers and fine-tuning these instead of the last layer(s)—currently the standard practice for fine-tuning in SSL. We verify this assumption in Table 5 and show that fine-tuning the most memorizing layers indeed yields the highest downstream performance on the fine-tuning dataset. This shows that localizing memorization might have practical application for more efficient fine-tuning in the future.

## 5 Unit-Level Localization of Memorization

Experiments from the previous section highlight that we are able to localize the memorization of data points in particular layers of the SSL encoders. This raises the even more fundamental question on

Table 5: **Fine-tuning most memorizing layers.** We train a ResNet9 encoder with SimCLR on CIFAR10 and fine-tune different (combinations of) layers on the STL10 dataset, resized to 32x32x3. We train a linear layer trained on top of the encoder (HEAD) and report STL10 test accuracy after fine-tuning. Fine-tuning the most memorizing layer(s), in contrast to the last layer(s), yields higher fine-tuning results.

| Fine-tuned Layers | Accuracy (%) ↑ |
|---|---|
| None (HEAD) | 48.6% ± 1.12% |
| 6 (highest $\Delta$LayerMem) + HEAD | **53.0% ± 0.86%** |
| 8 (last layer, highest LayerMem) + HEAD | 52.7% ± 0.97% |
| 6,8 + HEAD | **56.7% ± 0.84%** |
| 7,8 + HEAD | 55.3% ± 0.77% |
| 4,6,8 (highest $\Delta$LayerMem) + HEAD | **57.9% ± 0.79%** |
| 6,7,8 + HEAD | 56.5% ± 0.95% |

whether it is possible to trace down SSL memorization to a unit-level. To answer this question, we design UnitMem, a new metric to localize memorization in individual units of SSL encoders. We use the term *unit* to refer to both an activation map from a convolutional layer (single-layer output channel) or an individual neuron within a fully connected layer. Our UnitMem metric quantifies for every unit $u$ in the SSL encoder how much $u$ is sensitive to, *i.e.,* memorizes, any particular training data point. Therefore, UnitMem relates the *maximum unit activation* that occurs for a data point $x_k$ in the training data (sub)set $\mathcal{D}' \subseteq \mathcal{D}$ with the *mean unit activation* on all other data points in $\mathcal{D}' \setminus \{x_k\}$.

The design of UnitMem is inspired by the class selectivity metric defined for SL by [39]. Class selectivity was derived from selectivity indices commonly used in neuroscience [19, 8, 25] and quantifies a unit's discriminability between different classes. It was used as an indicator of good generalization in SL. We provide more background in Appendix D.1. To leverage ideas from class selectivity for identifying memorization, we integrate three fundamental changes in our metric in comparison to the class selectivity metric. While class selectivity is calculated on *classes* of the test set and relies on class *labels*, our UnitMem is (1) *label-agnostic* and (2) computed on individual data points from the *training dataset* to determine their memorization. Additionally, to account for SSL's strong reliance on augmentations, (3) we calculate UnitMem over the expectation on the augmentation set used during training. Research from the privacy community [49, 34] suggests that those augmentations leave a stronger signal in ML models than the original data point, *i.e.,* relying on the unaugmented point alone might under-report memorization. We verify this effect in Figure 5 in Appendix C.1. We note that through these fundamental changes UnitMem is able to measure memorization of *individual data points* within a class rather than to solely distinguish between classes or concepts like the original class selectivity. We provide further insights into this difference and perform experimental verification which highlights that UnitMem captures individual data points' memorization rather than capturing classes or concepts in Appendix C.2.

To formalize our UnitMem, we first define the mean activation $\mu$ of unit $u$ on a training point $x$ as

$$\mu_u(x) = \mathop{\mathbb{E}}_{x' \sim \mathrm{Aug}(x)} \mathrm{activation}_u(x'), \tag{5}$$

where the activation for convolutions feature maps is averaged across all elements of the feature map and for fully connected layers is an output from a single neuron (which is averaged across all patches of $x$ in ViTs). Further, for the unit $u$, we compute the maximum mean activation $\mu_{max,u}$ across all instances from $\mathcal{D}'$, where $N = |\mathcal{D}'|$, as

$$\mu_{max,u} = \max(\{\mu_u(x_i)\}_{i=1}^N). \tag{6}$$

Let $k$ be the index of the maximum mean activation $\mu_u(x_k)$, *i.e.,* the $argmax$. Then, we calculate the corresponding mean activity $\mu_{-max}$ across all the remaining $N-1$ instances from $\mathcal{D}'$ as

$$\mu_{-max,u} = \mathrm{mean}(\{\mu_u(x_i)\}_{i=1,i \neq k}^N). \tag{7}$$

Finally, we define the UnitMem of unit $u$ as

$$\mathrm{UnitMem}_{\mathcal{D}'}(u) = \frac{\mu_{max,u} - \mu_{-max,u}}{\mu_{max,u} + \mu_{-max,u}}. \tag{8}$$

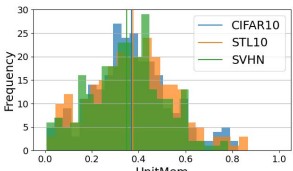
(a) **Different datasets.**

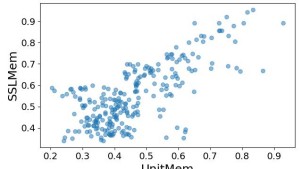
(b) `UnitMem` **vs** `SSLMem`.

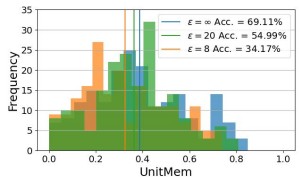
(c) `UnitMem` **with DP.**

Figure 1: **Insights into** `UnitMem`**.** We train a ResNet9 encoder with SimCLR: **(a)** Different datasets, including SVHN, CIFAR10, and STL10. We report the `UnitMem` of the last convolutional layer (conv4_2); **(b)** Comparing alignment between `SSLMem` and `UnitMem` on CIFAR10. Data points with higher general memorization (`SSLMem`) tend to experience higher `UnitMem`; **(c)** Using different strengths of privacy protection according to DP during training on CIFAR10 and Vit-Base

The value of the `UnitMem` metric is bounded between 0 and 1, where 0 indicates that the unit is equally activated by all training data points, while value 1 denotes exclusive memorization, where only a single data point triggers the activation, while all other points leave the unit inactive.

## 5.1 Experimental Results and Observations

We present our core results and provide detailed additional ablations on our `UnitMem` Appendix C.1.

**Highly Memorizing Units Occur over Entire Encoder.** Our analysis highlights that over all encoder architectures and SSL training frameworks, highly memorizing units are spread over the entire encoder. While, on average, earlier layers exhibit lower `UnitMem` than deeper layers, even the first layer contains highly memorizing units as shown in Figure 3 (first row). Figure 1a shows that this trend holds over different datasets. Yet, the SVHN dataset, which is visually less complex than the CIFAR10 or STL10 dataset, has the lowest number of highly memorizing units. This observation motivates us to study the relationship between the highest memorized (atypical or hard to learn) data points and the highest memorizing units.

**Most Memorized Samples and Units Align.** To draw a connection between data points and unit memorization, we analyze which data points are responsible for the highest $\mu_{max}$ scores. This corresponds to a data point which causes the highest activations of a unit, while other points activate the unit only to a small degree or not at all. We show the results in Figure 1b (also in Table 12 as well as in Figure 7 in Appendix C.1). For each unit $u$ in the last convolutional layer of the ResNet9 trained on CIFAR10, we measure its `UnitMem` score, then we identify which data point is responsible for the unit's $\mu_{max,u}$, and finally measure this point's `SSLMem` score. We plot the `UnitMem` and `SSLMem` scores for each unit and its corresponding point. Our results highlight that the data points that experience the highest memorization according to the `SSLMem` score are also the ones memorized in the most memorizing units. Given the strong memorization in individual units, we next look into two methods to reduce it and analyze their impact.

**Differential Privacy reduces Unit Memorization.** The gold standard to guarantee privacy in ML is Differential Privacy (DP) [21]. DP formalizes that any training data point should only have a negligible influence on the final trained ML model. To implement this, individual data points' gradients during training are clipped to a pre-defined norm, and controlled amounts of noise are added [1]. This limits the influence that each training data point can have on the final model. Building on the DP framework for SSL encoders [50], we train a ViT-Tiny using MAE on CIFAR10 with three different privacy levels—in DP usually indicated with $\varepsilon$. We train non-private ($\varepsilon = \infty$), little private ($\varepsilon = 20$), and highly private ($\varepsilon = 8$) encoders and apply our `UnitMem` to detect and localize memorization. Our results in Figure 1c highlight that while with increasing privacy levels, the average `UnitMem` decreases, there are still individual units that experience high memorization.

**Data Point vs Class Memorization.** Since stronger training augmentations yield higher class clustering [31] (*i.e.,* the fact that data points from the same downstream class are close to each other in representation space but distant to data points from other classes), we also analyze how the SSL encoders differ from the standard class discriminators, namely SL trained models. Therefore, we

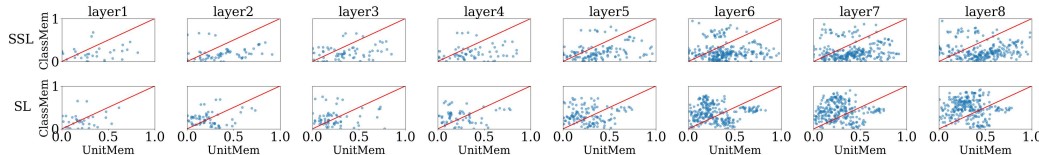

Figure 3: **Significantly more *(less)* units memorize data points rather than classes in SSL *(SL)*.** We measure the `ClassMem` vs `UnitMem` for 10000 samples from CIFAR100, with 100 random samples per class. Each i-th column represents the i-th convolutional layer in ResNet9, with 8 convolution layers, where the 1st row is for SSL while the 2nd row for SL. The red diagonal line denotes $y = x$.

go beyond our previous experiments that measure memorization of units with respect to individual data points and additionally study unit memorization at a class-granularity. Therefore, we adjust the class selectivity metric from [39] to perform on the training dataset rather than on the test data set as the original class selectivity. To avoid confusion between the two versions, refer to our adapted metric as `ClassMem` (see Appendix D.2 for an explicit definition). Equipped with `UnitMem` and `ClassMem`, we study the behavior of SSL encoders and compare between SSL and SL. For our comparison, we train an encoder with SimCLR and a model with SL using the standard cross entropy loss, both on the CIFAR100 dataset using ResNet9. We remove the classification layer from the SL trained model to obtain the same architecture as for the encoder trained with SimCLR.

For comparability, we early stop the SL training once it reaches a comparable performance to the linear probing accuracy on CI-FAR100 obtained by the SSL encoder. Our results in Figure 2 show that overall, in SSL throughout all layers, average memorization of individual data points is higher than class memorization, whereas in SL, in deeper layers, the class memorization increases significantly. We hypothesize that this effect is due to earlier layers in SL learning more general features which are independent of the class whereas later layers learn features that are highly class dependent. For SSL, such a difference over the network does not seem to exist; both

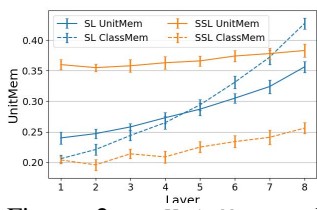

Figure 2: `UnitMem` **and** `ClassMem` **for SL and SSL.**

scores increase slightly, however, probably due to the SSL learning paradigm, memorization of individual data points remains higher.

To better understand the memorization of units on the micro level, we investigate further the individual units in each layer. In Figure 3, we plot the `ClassMem` vs `UnitMem` for each unit and in each of the eight encoder layers of ResNet9. Most units for SSL (row 1) constantly exhibit higher `UnitMem` than `ClassMem`, *i.e.,* they cluster under the diagonal line, which suggests that most units memorize individual data points across the whole network. Contrary, the initial layers for the model trained with SL have a slight tendency to memorize data points over entire classes whereas in later layers, this trend drastically reverses and most units memorize classes. In Appendix C.4, we investigate how this different memorization behavior between SL and SSL affects downstream generalization.

## 5.2 Verification of Unit-Based Memorization

We verify the unit-based localization of memorization with our `UnitMem` by deliberately inserting memorization of particular units and checking if our `UnitMem` correctly detects it. Therefore, we first train a SimSiam-based [17] ResNet18 encoder trained on the CIFAR10 dataset. We select SimSiam over SimCLR for this experiment since SimCLR, as a contrastive SSL framework, cannot train on a single data point. Then, using `LayerMem`, we identity the last convolutional layer in ResNet18 (*i.e.,* layer *4.1.conv2*) as the layer with the highest `LayerMem` and $\Delta$`LayerMem` memorization. We select the unit from the layer with the highest $\mu_{max}$ and also pick a unit with no activation ($\mu_{max} = 0$) for some test data points. Then, we fine-tune these units using a single test data point and report the change in `UnitMem` for the chosen units in Table 13. The results show that our `UnitMem` correctly detects the increase in memorization in both units. Additionally, we analyze the impact of zero-ing out the most or least memorizing vs. random units. Again with the argument that memorization is required for downstream generalization in SSL [47], we expect the highest performance drop when

Table 6: **Removing the least/most memorized units according to `UnitMem` preserves most/least linear probing performance**. We prune according to units with highest or lowest `UnitMem` either per layer or for the entire network (total). We also present baselines where we prune randomly selected units. The standard deviation for this baseline is reported over 10 independent trials where different random units were pruned. We train the ResNet9 encoder using CIFAR10 and compute the `UnitMem` score using 5000 data points from the train set.

| Pruning Strategy | % of Selected Units | Downstream Accuracy (%) | | |
|---|---|---|---|---|
| | | CIFAR10 | SVHN | STL10 |
| No Pruning | - | 70.44 | 78.22 | 69.12 |
| Top `UnitMem` per layer | 10 | 53.04 | 63.84 | 50.94 |
| Random per layer | 10 | $58.09 \pm 1.76$ | $67.04 \pm 2.44$ | $55.71 \pm 2.18$ |
| Low `UnitMem` per layer | 10 | 62.58 | 72.26 | 59.26 |
| Top `UnitMem` per layer | 20 | 48.30 | 55.88 | 43.18 |
| Random per layer | 20 | $51.34 \pm 1.21$ | $58.01 \pm 1.34$ | $46.74 \pm 0.97$ |
| Low `UnitMem` per layer | 20 | 54.84 | 62.60 | 50.02 |
| Top `UnitMem` total | 10 | 49.16 | 61.28 | 47.30 |
| Random total | 10 | $56.77 \pm 2.09$ | $67.09 \pm 1.56$ | $53.89 \pm 2.33$ |
| Low `UnitMem` total | 10 | 62.62 | 72.28 | 59.30 |

zero-ing out the most memorizing units. Our results in Table 6 in and in Appendix C.1 confirm this hypothesis and show that removing the most memorizing units yields the highest loss in linear probing accuracy on various downstream tasks while pruning the least memorized units preserves better downstream performance than removing random units. These results suggest that future work may benefit from using our `UnitMem` metric for finding which units within a network can be pruned while preserving high performance.

# 6 Conclusions

We propose the first practical metrics for localizing memorization within SSL encoders on a per-layer and per-unit level. By analyzing different SSL architectures, frameworks, and datasets using our metrics, we find that while memorization in SSL increases in deeper layers, a significant fraction of highly memorizing units can be encountered over the entire encoder. Our results also show that SSL encoders significantly differ from SL trained models in their memorization patterns, with the former constantly memorizing data points and the latter increasingly memorizing classes. Finally, using our metrics for localizing memorization presents itself as an interesting direction towards more efficient encoder fine-tuning and pruning.

### Acknowledgments

The project on which this paper is based was funded by the Deutsche Forschungsgemeinschaft (DFG, German Research Foundation), Project number 550224287. Additional funding came from the Initiative and Networking Fund of the Helmholtz Association in the framework of the Helmholtz AI project call under the name „PAFMIM", funding number ZT-I-PF-5-227. Responsibility for the content of this publication lies with the authors.

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

# A  Glossary

For the reader's convenience, we provide a glossary with all important terms and concepts related to our work in Table 7.

Table 7: **Glossary.** We present a concise overview on the concepts relevant to this work.

| Concept | Explanation |
|---|---|
| **Atypical examples** | Data points that are uncommon in the data distribution and different in terms of their features. Examples: Figure 1 from [47]. Sometimes also called "outliers". |
| **Class Selectivity** | A metric proposed by [39] which quantifies a unit's discriminability between different classes, measured on the test data. |
| **ClassMem** | Our adaptation of Class Selectivity measured on the training data. |
| **Downstream Generalization** | Expresses how well an encoder is suited to solve some downstream tasks. For classification, it is often measured by linear probing, i.e., training an additional classification layer on top of the encoder output. |
| **LayerMem** | Our proposed metric to quantify memorization of any layer in the SSL encoder. |
| **Memorization** | A phenomenon where a machine learning model stores detailed information on its training data. |
| **Memorized Data Point** | A data point that experiences high memorization by a machine learning model. |
| **Memorization Pattern** | A general trend in the low-level memorization of an SSL encoder, i.e., in which layers or units do memorization localize. |
| **Unit** | Term used to refer to an individual neuron in fully connected layers or a channel in convolutional layers. |
| **UnitMem** | Our proposed metric to quantify the memorization of any unit in the SSL encoder. |

# B  Experimental Setup

**Datasets.**  We base our experiments on ImageNet ILSVRC-2012 [42], CIFAR10 [32], CIFAR100 [32], SVHN [40], and STL10 [18].

**Models.**  We use the ResNet family of models [28], including ResNet9, ResNet18, ResNet34, and ResNet50. In Table 8, we present the detailed architecture of the ResNet9 model.

Table 8: **Architecture of ResNet9.** In the *Number of Units* column, we present the number of activation maps (corresponding to individual filters in the filter bank).

| Conv-Layer ID | Layer Name | Number of Units | Number of Parameters |
|---|---|---|---|
| 1 | Conv1 | 32 | 896 |
| - | BN1 | 32 | 64 |
| - | MaxPool1 | 32 | 0 |
| 2 | Conv2-0 | 64 | 18496 |
| - | BN2-0 | 64 | 128 |
| - | MaxPool2-0 | 64 | 0 |
| 3 | Conv2-1 | 64 | 36928 |
| - | BN2-1 | 64 | 128 |
| - | MaxPool2-1 | 64 | 0 |
| 4 | Conv2-2 | 64 | 36928 |
| - | BN2-2 | 64 | 128 |
| - | MaxPool2-2 | 64 | 0 |
| 5 | Conv3 | 128 | 73856 |
| - | BN3 | 128 | 256 |
| - | MaxPool3 | 128 | 0 |
| 6 | Conv4-0 | 256 | 295168 |
| - | BN4-0 | 256 | 512 |
| - | MaxPool4-0 | 256 | 0 |
| 7 | Conv4-1 | 256 | 590080 |
| - | BN4-1 | 256 | 512 |
| - | MaxPool4-1 | 256 | 0 |
| 8 | Conv4-2 | 256 | 590080 |
| - | BN4-2 | 256 | 512 |
| - | MaxPool4-2 | 256 | 0 |

**SSL Frameworks.**  We base our experimentation on four state-of-art SSL encoders: MAE [30], SimCLR [16], DINO [13], and SimSiam [17].

**Training Setup.**   Our experimental setup for training the encoders mainly follows [47] and we rely on their naming conventions and refer to the data points that are used to train encoder $f$, but not reference encoder $g$ as *candidate* data points. In total, we use 50000 data points as training samples for CIFAR10, SVHN, and STL10 and 100000 for ImageNet with 5000 candidate data points per dataset. The encoders evaluated in the paper are trained with batch size 1024, and trained 600 epochs for CIFAR10, SVHN, and STL10, and 300 epochs for ImageNet. We set the batch size to 1024 for all our experiments and train for 600 epochs on CIFAR10, SVHN, and STL10, and for 300 epochs on ImageNet. As a distance metric to measure representation alignment, we use the $\ell_2$ distance. We repeat all experiments with three independent seeds and report average and standard deviation. For reproducibility, we detail our full setup in Table 9 with the standard parameters that are used throughout the paper if not explicitly specified otherwise.

**Training Augmentations.**   We generate augmentations at random from the following augmentation sets (p indicates augmentation probability):

- **SL, standard, (referred to as *weak augmentations*):** ColorJitter(0.9-0.9-0.9-0.5, p=0.4), RandomHorizontalFlip(p=0.5), RandomGrayscale(p=0.1), RandomResizedCrop(size=32)

- **SSL, standard, (referred to as *normal augmentations*):** ColorJitter(0.8-0.8-0.8-0.2, p=0.8), RandomHorizontalFlip(p=1.0), RandomGrayscale(p=0.2), RandomResizedCrop(size=32)

- **SSL, stronger, (referred to as *strong augmentations*):** ColorJitter(0.8-0.8-0.8-0.2, p=0.9), RandomHorizontalFlip(p=1.0), RandomGrayscale(p=0.5), RandomResizedCrop(size=32), RandomVerticalFlip(p=1.0)

- **SSL (independent):** GaussianBlur(kernel_size=(4, 4), sigma=(0.1, 5.0), p=0.8), RandomInvert(p=0.2), RandomResizedCrop(size=32), RandomVerticalFlip(p=1.0)

- **Masking (MAE):** 75% random masking

**Details on Computing** `UnitMem`**.**   Relying on insights from [33], we calculate Equation (5) for the activations within `UnitMem` over ten augmentations since this has shown to yield a strong signal on the augmented data point. For convolutional feature maps, the activation of the unit is calculated as the average of all elements in the feature map. In ViTs, where we measure activation over fully-connected layers, we compute the activation per neuron and average across all patches of a given input. For example, for ViT Tiny encoder pretrained on CIFAR10, the input image of resolution 32x32 is *patchified* into 64 patches, each of size 4x4. Then, each patch is represented by a 192 dimensional embedding. The classification (CLS) embedding is prepended to the remaining 64 embeddings. Overall, we obtain 65 patches. The last fully connected layer has 192 neurons. For each neuron, we average its activations across the 65 patches. In the case of ViT Base, we have 768-dimensional embeddings and 197 patches for the input image of resolution 224x224.

**Details on Fine-Tuning with one Test Data Point**   We provide the exact details of our experiments to verify our `UnitMem` through deliberate insertion of memorization in Section 5.2. We train a SimSiam-based [17] ResNet18 encoder on the CIFAR10 dataset and use `LayerMem` to identity layer *4.1.conv2*, *i.e.,* the last convolutional layer in ResNet18, as the layer with highest accumulated memorization. We select the unit from the layer with the highest $\mu_{max}$ and also pick a unit with no activation ($\mu_{max} = 0$) for some test data points. Then, for compatability with pytorch which does not support individual unit training, we lock all parameters except for the targeted *layer* and train the model with a single sample from the testing dataset. We choose the sample that achieves the highest activation $\mu_{max}$ on the unit with the highest `UnitMem`. We save the checkpoints after each epoch and test the $\mu_{max}$ for the selected two units. Our results in Table 13 show that the value of $\mu_{max}$ for the selected data point increases in both units and the data point remains the one responsible for the $\mu_{max}$.

**Details on Hardware resources usage**   We finish all our experiments on two devices: a cloud server with four A100 GPUs and a local workstation with Intel 13700k CPU, Nvidia 4090 graphics card and 64GB of RAM

Table 9: **Training Setup for SSL Frameworks and Hyperparameters.** Two numbers denote ImageNet / Others.

| | Model Training | | | | Linear Probing | | | |
|---|---|---|---|---|---|---|---|---|
| | MAE | SimCLR | DINO | SimSiam | MAE | SimCLR | DINO | SimSiam |
| Training Epochs | 300 / 600 | 300 / 600 | 300 / 600 | - / 200 | 45 / 90 | 45 / 90 | 45 / 90 | - / 30 |
| Warm-up Epochs | 30 / 60 | 30 / 60 | 30 / 60 | - / 24 | 5 / 10 | 5 / 10 | 5 / 10 | - / 3 |
| Batch Size | 2048 | 4096 | 1024 | 128 | 4096 | 4096 | 4096 | 256 |
| Optimizer | AdamW | LARS | AdamW | SGD | LARS | LARS | LARS | SGD |
| Learning rate | 1.2e-3 | 4.8 | 2e-3 | 2.5e-2 | 1.6 | 4.8 | 1.6 | 5e-2 |
| Learning rate Schedule | Cos. Decay | Cos. Decay | Cos. Decay | Cos. Decay | Cos. Decay | Cos. Decay | Cos. Decay | Cos. Decay |

# C  Additional Experiments

## C.1  Additional Insights into `UnitMem`

`UnitMem` **increases over training.** First, we assess how `UnitMem` evolves over training of the SSL encoder. Therefore, we train a ResNet9 encoder using SimCLR on the CIFAR10 dataset for 800 epochs, using 120 warm-up epochs. Every five epochs, we measure the `UnitMem`. Our results in Figure 4 depict the average `UnitMem` of the ResNet9's last convolutional layer.

We observe that the unit memorization monotonically increases throughout training and that the increase is particularly high during the first epochs. After the warm-up, we observe that the increase in unit memorization stagnates until the level of memorization on the unit level converges. The same trend can be observed over all layers which indicates that SSL encoders increase unit memorization throughout training.

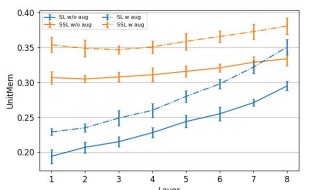

Figure 4: **Average** `UnitMem` **of layer 8 over training.**

**Measuring** `UnitMem` **without using augmentations leads to an under-reporting of memorization.**

To assess the impact on using augmentation to implement Equation (5) for the calculation of our `UnitMem` has an impact on the reported results, we train two ResNet9 models on the CIFAR100 dataset, one using SimCLR, the other one using standard SL with cross entropy loss.

During training we rely on the standard augmentations for SL and SSL reported above. To measure memorization, we once use ten augmentations from the training augmentation set, and no augmentations otherwise and report the results in Figure 5. We find that while the trend of the reported memorization is equal in both settings, the `UnitMem` measured without augmentations remains constantly lower than when measured with augmentations. This suggests that when measuring `UnitMem`, it is important to use augmentations to avoid under-reporting of the memorization.

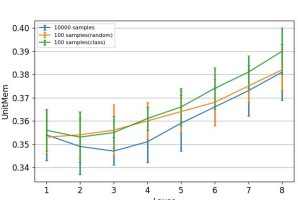

Figure 5: `UnitMem` **w & w/o augmentations.**

**The number of data points used to measure** `UnitMem` **does not have a significant impact on the reported memorization.** Using the same ResNet9, trained with SimCLR on CIFAR100, we assess whether the number of data points that we use to calculate `UnitMem` (the size of $\mathcal{D}'$) has an impact on the reported memorization. Then, we measure `UnitMem` using 100 random data point chosen one from each class in CIFAR100, 100 purely randomly chosen data points, and randomly chosen CIFAR100 data points. We present our findings in Figure 6. Our results highlight that all the lines are within each other's standard deviation, indicating that there is no significant difference in the reported `UnitMem`, dependent on the make up of the dataset $\mathcal{D}'$.

Figure 6: **Size of** $\mathcal{D}'$.

**Most memorized data points align with the most memorizing units.** We train a ResNet9 on CIFAR10 using SimCLR and measure `UnitMem` for the 300 most and 300 least memorized data points identified using `SSLMem` by [47]. The measurement of the two sets (most vs least memorized

data points) is performed independently. Our results in Figure 7 show that the `UnitMem` calculated on the most memorized data points is significantly higher than on the least memorized data points (we verify the significance with a statistical $t$-test in Table 11.

While also some of the least memorized data points lead to a high activation of the units, highest activation (on average and in particular) can be observed for the most memorized data points. This underlines the trend observed in Table 12 which shows that highly memorized data points align with the highly memorizing units.

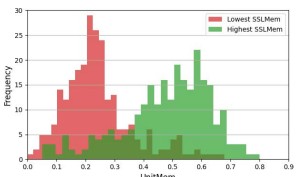

Figure 7: **Least vs most memorized data points.**

**Computing `UnitMem` based on the median yields similar results to using the mean.** Our `UnitMem` metric is inspired by the class selectivity defined for SL by [39] which quantifies a unit's discriminability between different classes, see Appendix D.1. Yet, we calculate the $\mu_{-max,u}$ in Equation (7) using the *median* on the other individual training data points' activations while `ClassSelectivity` computes their equivalent of $\mu_{-max,u}$ using the *mean* on all other test classes' activations.

We show in Figure 8 over the 300 most and least memorized data points for a ResNet9 trained with SimCLR on CIFAR10 that using the median for `UnitMem` yields very similar results to using the mean.

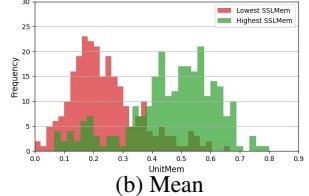

(a) Median    (b) Mean

Figure 8: **Mean vs Median.**

**For SSL, the concrete augmentation set has no strong impact when measuring `UnitMem`.** We additionally set out to study the impact of the augmentation set used to calculate `UnitMem`. Therefore, we calculate `UnitMem` on the ResNet9 trained on CIFAR10 using SimCLR using different augmentations sets. For SSL, we measure once with the standard training augmentations ("Normal"), with an independent set of augmentations of similar strength ("Independent"), with a weaker augmentation set for which we rely on the augmentations used to train the SL model ("Weak"), and an independent very strong set of augmentations modeled after MAE training and using a masking of 75% of the input image ("Masking"). Our results in Figure 9 depict the `UnitMem` over the last convolutional layer of the ResNet9 encoders.

They highlight that the weak and independent augmentations report extremely similar `UnitMem` to the original set of training augmentations used. For SL, the impact of using different augmentations during training and measuring `UnitMem` is more expressed. We also measured for a weak augmentation set ("Normal"), an independent weak set ("Independent"),

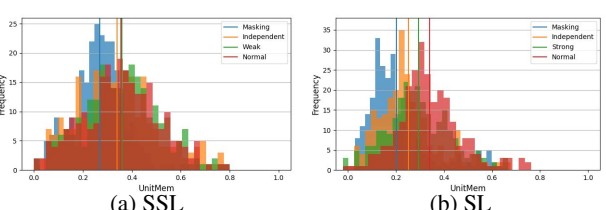

(a) SSL    (b) SL

Figure 9: **Different augmentation sets.**

strong augentations for which we relied on the standard SSL augmentations ("Strong"), and the 75% masking ("Masking"). We observe that using the augmentations from training to calculate `UnitMem` yields the highest localization of memorization.

**Stronger augmentations reduce memorization.** We also analyze how the training augmentation strength can impact the final encoder's `UnitMem`. We use ColorJitter, HorizontalFlip, RandomGrayscale, and RandomResizedCrop as augmentations. Their strength is determined by the probability of applying them and their level of distortion. In Appendix B, we present the exact parameters specified for each of them under different strengths. Our results in Figure 10 suggest that stronger augmentations yield lower per-unit memorization. These findings are in line with prior theoretical

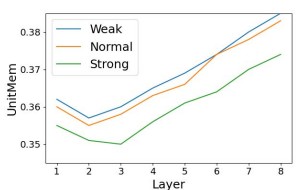

Figure 10: **Different augmentation sets.**

work on SSL [48] highlighting that SSL performs foremost the task of instance discrimination (*i.e.,* differentiating between individual images), but achieves clustering according to classes due to the augmentations: with stronger augmentations, multiple data points' augmented views will look extremely similar (*e.g.,* the wheels of two different images of cars), such that they eventually activate the same unit. Thereby, this unit memorizes individual data points less while units trained with weaker augmentations depend on and memorize individual data points more. Note that we do not observe a strong dependency of our reported `UnitMem` on the concrete augmentation set used to calculate the metric (see Equation (5)) as we show in Figure 9 in Appendix C.1. Yet, using the original set of training augmentations, as we do for our experiments, yields the strongest signal.

**Stronger weight decay reduces memorization.** To analyze how training weight decay affects the final encoder's `UnitMem`, we train a ResNet9 using SimCLR on CIFAR10 with three different levels of weight decay. Our results in Figure 11 show that stronger weight decay yields lower memorization, yet also decreases linear probing accuracy.

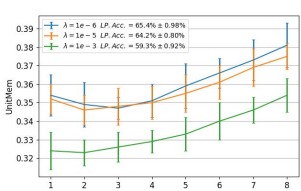

**Different SSL frameworks yield similar memorization pattern.** We compare the `UnitMem` score between corresponding layers of a ResNet50 pre-trained on ImageNet with SimCLR and DINO, as

Figure 11: Different weight decay

well as for ViT-Base encoders pre-trained on ImageNet with DINO and MAE. We ensure that the number of epochs, batch sizes, training dataset sizes, and the resulting linear probing accuracies of the encoders are similar for direct comparability. Our results in Figure 12 depict the `UnitMem` of the last convolutional layer of the ResNet50, and the final block's fully-connected layer in the ViT. The plot indicates that the different SSL frameworks applied to the same architecture with the same dataset yield similar memorization pattern.

Table 10: **The memorization in ViT occurs primarily in the deeper blocks and more in the fully-connected than attention layers.** We present the results for ViT Tiny pre-trained on CIFAR10 using MAE.
$\Delta\texttt{LayerMem}_N^{ATT} = \texttt{LayerMem}_N^{ATT} - \texttt{ResBlock}_{N-1}^{FC}$, $\Delta\texttt{LayerMem}_N^{FC} = \texttt{LayerMem}_N^{FC} - \texttt{ResBlock}_N^{ATT}$, $\Delta\texttt{BlockMem}_N = \texttt{ResBlock}_N^{FC} - \texttt{ResBlock}_{N-1}^{FC}$.

| ViT Block | Attention Layer | | | Fully-Connected Layer | | | |
|---|---|---|---|---|---|---|---|
| Number | LayerMem | $\Delta$LayerMem | ResBlock | LayerMem | $\Delta$LayerMem | ResBlock | $\Delta$BlockMem |
| 1 | 0.006 | - | 0.007 | 0.020 | - | 0.022 | - |
| 2 | 0.028 | 0.006 | 0.028 | 0.039 | 0.011 | 0.040 | 0.018 |
| 3 | 0.046 | 0.006 | 0.047 | 0.060 | 0.013 | 0.061 | 0.021 |
| 4 | 0.067 | 0.006 | 0.067 | 0.083 | 0.017 | 0.085 | 0.024 |
| 5 | 0.092 | 0.007 | 0.091 | 0.105 | 0.014 | 0.106 | 0.021 |
| 6 | 0.114 | 0.008 | 0.114 | 0.129 | 0.015 | 0.131 | 0.025 |
| 7 | 0.140 | 0.009 | 0.139 | 0.155 | 0.016 | 0.156 | 0.025 |
| 8 | 0.164 | 0.008 | 0.164 | 0.182 | 0.018 | 0.182 | 0.026 |
| 9 | 0.191 | 0.009 | 0.190 | 0.210 | 0.020 | 0.211 | 0.029 |
| 10 | 0.220 | 0.009 | 0.220 | 0.240 | 0.020 | 0.241 | 0.030 |
| 11 | 0.249 | 0.008 | 0.249 | 0.271 | 0.022 | 0.271 | 0.030 |
| 12 | 0.280 | 0.009 | 0.280 | 0.303 | 0.023 | 0.304 | 0.033 |

**Additional Verification of `UnitMem`.** We present the additional verification of the `UnitMem` metric in Table 14. Therein, we perform two additional experiments to the verification presented in Section 5.2. First, we fine-tune the most memorizing unit and the inactive unit with 300 (instead of 1) data points from the test set (a). We observe that the data points that expe-

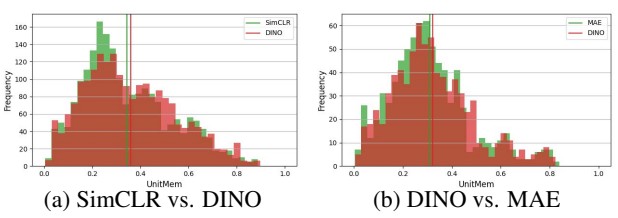

(a) SimCLR vs. DINO    (b) DINO vs. MAE

Figure 12: **Different SSL frameworks.**

rienced the highest memorization for the selected unit remains the highest memorized of the 300 data points. Additionally, it experiences the highest memorization in the unit that used to be inactive. Second, we fine-tune the most memorizing unit and the inactive unit with the most memorized data point, but with a batch-size of 300 were we duplicate the data point 300 times (b). We observe that

Table 11: **Most vs Least Memorized Data Points**. We train a ResNet9 using SimCLR on CIFAR10 follwoing the setup by [47]. We then take the 50 most and 50 least memorized data points according to `SSLMem` and calculate the `UnitMem` over for the two sets of points. In the table, we report the average per-layer `UnitMem` of the two sets independently. We also perform a statistical $t$-test to find whether the `UnitMem` scores differ among most and least memorized data points. With $p << 0.05$, we are able to reject the null-hypothesis and find that the memorization according to `UnitMem` differs significantly between the most and least memorized data points.

| Layer Name | mean `UnitMem` most memorized (10% units) | mean `UnitMem` least memorized (10% units) | t-test p-value |
|---|---|---|---|
| conv1 | 0.507 | 0.235 | 65.89/0.00 |
| conv2-0 | 0.501 | 0.231 | 66.25/0.00 |
| conv2-1 | 0.503 | 0.233 | 65.94/0.00 |
| conv2-2 | 0.512 | 0.240 | 65.12/0.00 |
| conv3 | 0.509 | 0.242 | 64.13/0.00 |
| conv4-0 | 0.514 | 0.246 | 63.67/0.00 |
| conv4-1 | 0.515 | 0.245 | 64.09/0.00 |
| conv4-2 | 0.522 | 0.248 | 64.18/0.00 |

Table 12: **Highly memorized data points align with most memorizing units.** We select 10% of the most memorizing units according to `UnitMem` in the last layer (*conv-4-2*) of the ResNet9 encoder pre-trained on CIFAR10. The 1st row represents the number of times a given data point was responsible for $\mu_{max}$, the 2nd row counts for how many daat points this applies. The last column shows that the highest memorized sample (`SSLMem` of 0.891) is responsible for the $\mu_{max}$ in the largest number of units (5).

| Metric Used→ | *Average SSLMem Score* | | | | |
|---|---|---|---|---|---|
| Frequency↓ | 0.694 | 0.813 | 0.833 | 0.857 | 0.891 |
| # of times Responsible for $\mu_{max}$ | 1 | 2 | 3 | 4 | 5 |
| # of Samples | 10 | 2 | 1 | 1 | 1 |

the effect of the fine-tuning on this point's memorization is far more expressed than when fine-tuning with 300 different data points.

**Additional Verification of** `UnitMem`. In Table 13, we prune, *i.e.,* zero out neurons according to their level of memorization. Our results indicate that by pruning the most memorizing neurons, we cause the highest drop in downstream performance.

## C.2 `UnitMem` **Measures Memorization of Individual Data Points**

To highlight that `UnitMem` reports memorization of individual data points rather than the a unit's ability to recognize class-wide concepts, we designed an additional experiment. For the experiment, we rely on the class concept of "wheel" as an example. In the STL10 dataset, three classes have a concept wheel, namely Truck, Plane, Car. If `UnitMem` was to report simply a unit's sensitivity to concepts of different classes (rather than individual data points), we would see a drop in `UnitMem` as we increase the percentage of data points with the concept wheel in the batch used to compute the metric. This is because then all data points should equally activate the unit, resulting in low memorization according to Equation (8).

We perform the experiment in Table 34 with 1000 data points chosen from different classes, namely 1) all classes (here 30% of the data points have wheels), 2) the classes Truck, Plane, Car (close to 100% of the samples now have the concept of wheels), and 3) purely the class car (close to 100% wheels). In 2) and 3), close to 100% of the samples now have the concept of wheels. Thus, if the units were responsible for the concept wheel, they would have a very high activation over all samples and the reported `UnitMem` should be very low. However, in our results, we see that we do have units with very high `UnitMem`. These can, in turn not be the units for the class-concept wheel, but must be units that focus on individual characteristics of the individual training images. This means that there must be unique features from the individual images that are still memorized that go beyond the concepts that are the same within a class.

Table 13: **Verification of the `UnitMem` metric for memorization in individual units.** The SSL model based on SimSiam with ResNet18 architecture and trained on CIFAR10 is fine-tuned on a single data point. We select two units with the highest and lowest `UnitMem` scores. The data point used for fine-tuning achieves $\mu_{max}$ in both units. The `UnitMem` score increases only for the two selected units while it remains unchanged for the remaining units.

| Targeted | *Number of Fine-Tuning Epochs* | | | | | |
|---|---|---|---|---|---|---|
| Unit | 0 | 10 | 50 | 200 | 500 | 1000 |
| Highest `UnitMem` | 0.754 | 0.761 | 0.792 | 0.814 | 0.824 | 0.826 |
| Lowest `UnitMem` | 0 | 0 | 0 | 0.0008 | 0.0021 | 0.0109 |

Table 14: **The $\mu_{max}$ and $\mu_{min}$ after fine-tuning for different number of epochs.**

(a) **The $\mu_{max}$ and $\mu_{min}$ after fine-tuning for different numbers of epochs.** This is fine-tuned with 300 data samples from the test dataset. The samples were not seen during the initial training of the encoder, thus only a single filter is affected by them.

| trained nodes | original | 10 epoch | 50 epoch | 200 epoch | 500 epoch | 1000 epoch |
|---|---|---|---|---|---|---|
| Most Mem filter $77^{th}$ | 0.754 | 0.766 | 0.809 | 0.819 | 0.826 | 0.828 |
| Least Mem filter $459^{th}$ | 0 | 0 | 0.011 | 0.038 | 0.046 | 0.051 |

(b) **The $\mu_{max}$ and $\mu_{min}$ after fine-tuning for different number of epochs** This is fine-tuned with only highest $\mu_{max}$ samples while 300 duplication from training datasets.

| trained nodes | original | 10 epoch | 50 epoch | 200 epoch | 500 epoch | 1000 epoch |
|---|---|---|---|---|---|---|
| Most Mem filter $77^{th}$ | 0.754 | 0.798 | 0.846 | 0.857 | 0.861 | 0.862 |
| Least Mem filter $459^{th}$ | 0 | 0.039 | 0.065 | 0.079 | 0.081 | 0.081 |

## C.3 Additional Insights into `LayerMem`

`LayerMem` **is not sensitive to the size and composition of the batch.** In our ablation study in , we show that `LayerMem` is not sensitive to the size and composition of batch it is computed on. The results can be found in Table 19, where report the `LayerMem` measured for different number of candidate data points. We pre-trained a ResNet9 using SimCLR on CIFAR10 and determined `LayerMem` on batches of different sizes. For each batch size, we use 3 independent seeds (*i.e.,* different batch compositions) and report the average `LayerMem` score and its standard deviation. The results show that the reported `LayerMem` score is, indeed, similar across all setups. This indicates `LayerMem` 's insensitivity to the choice of the batch used to compute it.

**Full Results with Memorization Scores over all Layers.** We present the `LayerMem` score for ResNet18 in Table 15, ResNet34 in Table 16, and ResNet50 in Table 17, all trained on CIFAR10 and using the SimCLR framework.

We show the further breakdown of the memorization within the layers in Table 18. We observe that the batch normalization layers (denoted as BN) together with the MaxPool layers have a negligible impact on memorization and most of the memorization in each layer is due to the convolutional operations. This is due to the much larger number of parameters in the convolutional filters than in the batch normalization layers and no additional parameters in the MaxPool layers, as shown in Table 8. However, the memorization reported per convolutional layer is not correlated with the number of parameters of the layer. For instance, our $\Delta$`LayerMem` reports the highest memorization for the 6-th layer, while layers 7 and 8 have each twice as many parameters, see Table 1.

Table 15: **Full results ResNet18.** We depict our `LayerMem` of the final trained model (at the end of training with CIFAR10, ResNet18 with SimCLR).

| Layer | LayerMem |
|---|---|
| conv1 | $0.074 \pm 0.010$ |
| max pool | $0.092 \pm 0.007$ |
| conv2-1 | $0.101 \pm 0.012$ |
| conv2-2 | $0.110 \pm 0.006$ |
| conv2-3 | $0.123 \pm 0.013$ |
| conv2-4 | $0.134 \pm 0.010$ |
| conv3-1 | $0.146 \pm 0.008$ |
| conv3-2 | $0.155 \pm 0.013$ |
| conv3-3 | $0.166 \pm 0.011$ |
| conv3-4 | $0.183 \pm 0.007$ |
| conv4-1 | $0.193 \pm 0.006$ |
| conv4-2 | $0.206 \pm 0.009$ |
| conv4-3 | $0.220 \pm 0.011$ |
| conv4-4 | $0.239 \pm 0.010$ |
| conv5-1 | $0.246 \pm 0.014$ |
| conv5-2 | $0.257 \pm 0.007$ |
| conv5-3 | $0.272 \pm 0.011$ |
| conv5-4 | $0.295 \pm 0.012$ |
| averge-pool | $0.266 \pm 0.009$ |
| fully-connected | $0.224 \pm 0.010$ |
| softmax | $0.207 \pm 0.007$ |

Table 16: **Full results ResNet34.** We depict our `LayerMem` of the final trained model (at the end of training with CIFAR10, ResNet34 with SimCLR).

| Layer | LayerMem |
|---|---|
| conv1 | $0.037 \pm 0.008$ |
| max pool | $0.069 \pm 0.013$ |
| conv2-1 | $0.078 \pm 0.008$ |
| conv2-2 | $0.083 \pm 0.007$ |
| conv2-3 | $0.091 \pm 0.012$ |
| conv2-4 | $0.096 \pm 0.009$ |
| conv2-5 | $0.107 \pm 0.016$ |
| conv2-6 | $0.115 \pm 0.010$ |
| conv3-1 | $0.124 \pm 0.011$ |
| conv3-2 | $0.128 \pm 0.013$ |
| conv3-3 | $0.131 \pm 0.007$ |
| conv3-4 | $0.138 \pm 0.008$ |
| conv3-5 | $0.143 \pm 0.013$ |
| conv3-6 | $0.149 \pm 0.015$ |
| conv3-7 | $0.157 \pm 0.013$ |
| conv3-8 | $0.166 \pm 0.009$ |
| conv4-1 | $0.172 \pm 0.006$ |
| conv4-2 | $0.178 \pm 0.010$ |
| conv4-3 | $0.181 \pm 0.012$ |
| conv4-4 | $0.186 \pm 0.008$ |
| conv4-5 | $0.194 \pm 0.013$ |
| conv4-6 | $0.201 \pm 0.007$ |
| conv4-7 | $0.205 \pm 0.009$ |
| conv4-8 | $0.211 \pm 0.011$ |
| conv4-9 | $0.218 \pm 0.006$ |
| conv4-10 | $0.227 \pm 0.012$ |
| conv4-11 | $0.235 \pm 0.010$ |
| conv4-12 | $0.246 \pm 0.007$ |
| conv5-1 | $0.257 \pm 0.011$ |
| conv5-2 | $0.264 \pm 0.014$ |
| conv5-3 | $0.273 \pm 0.008$ |
| conv5-4 | $0.285 \pm 0.012$ |
| conv5-5 | $0.299 \pm 0.011$ |
| conv5-6 | $0.313 \pm 0.015$ |
| averge-pool | $0.297 \pm 0.009$ |
| fully-connected | $0.241 \pm 0.013$ |
| softmax | $0.233 \pm 0.006$ |

**Ablation on `LayerMem`'s sensitivity.** We perform an additional ablation to show that `LayerMem` is not sensitive to the number of samples in the batch used to compute it or the composition of the batch, *i.e.,* which samples are chosen. Our results in Table 19 highlight that over different batches with 100, 500, 1000, and 5000 samples, the observed `LayerMem` scores are alike. This indicates `LayerMem` 's insensitivity to the choice of the batch used to compute it.

Table 17: **Full results ResNet50.** We depict our `LayerMem` of the final trained model (at the end of training with CIFAR10, ResNet50 with SimCLR).

| Layer | LayerMem |
|---|---|
| conv1 | $0.046 \pm 0.006$ |
| max pool | $0.066 \pm 0.012$ |
| conv2-1 | $0.071 \pm 0.008$ |
| conv2-2 | $0.068 \pm 0.013$ |
| conv2-3 | $0.073 \pm 0.012$ |
| conv2-4 | $0.079 \pm 0.015$ |
| conv2-5 | $0.082 \pm 0.014$ |
| conv2-6 | $0.083 \pm 0.010$ |
| conv2-7 | $0.088 \pm 0.007$ |
| conv2-8 | $0.094 \pm 0.011$ |
| conv2-9 | $0.103 \pm 0.014$ |
| conv3-1 | $0.109 \pm 0.010$ |
| conv3-2 | $0.112 \pm 0.012$ |
| conv3-3 | $0.118 \pm 0.009$ |
| conv3-4 | $0.123 \pm 0.007$ |
| conv3-5 | $0.127 \pm 0.010$ |
| conv3-6 | $0.133 \pm 0.011$ |
| conv3-7 | $0.136 \pm 0.013$ |
| conv3-8 | $0.140 \pm 0.008$ |
| conv3-9 | $0.144 \pm 0.005$ |
| conv3-10 | $0.149 \pm 0.008$ |
| conv3-11 | $0.156 \pm 0.011$ |
| conv3-12 | $0.165 \pm 0.007$ |
| conv4-1 | $0.168 \pm 0.012$ |
| conv4-2 | $0.175 \pm 0.010$ |
| conv4-3 | $0.181 \pm 0.006$ |
| conv4-4 | $0.187 \pm 0.009$ |
| conv4-5 | $0.192 \pm 0.008$ |
| conv4-6 | $0.198 \pm 0.014$ |
| conv4-7 | $0.204 \pm 0.010$ |
| conv4-8 | $0.211 \pm 0.008$ |
| conv4-9 | $0.217 \pm 0.011$ |
| conv4-10 | $0.225 \pm 0.005$ |
| conv4-11 | $0.231 \pm 0.015$ |
| conv4-12 | $0.235 \pm 0.011$ |
| conv4-13 | $0.241 \pm 0.012$ |
| conv4-14 | $0.248 \pm 0.009$ |
| conv4-15 | $0.253 \pm 0.011$ |
| conv4-16 | $0.262 \pm 0.016$ |
| conv4-17 | $0.268 \pm 0.012$ |
| conv4-18 | $0.279 \pm 0.011$ |
| conv5-1 | $0.292 \pm 0.008$ |
| conv5-2 | $0.293 \pm 0.005$ |
| conv5-3 | $0.298 \pm 0.012$ |
| conv5-4 | $0.308 \pm 0.010$ |
| conv5-5 | $0.316 \pm 0.014$ |
| conv5-6 | $0.315 \pm 0.012$ |
| conv5-7 | $0.326 \pm 0.007$ |
| conv5-8 | $0.327 \pm 0.011$ |
| conv5-9 | $0.335 \pm 0.013$ |
| averge-pool | $0.328 \pm 0.007$ |
| fully-connected | $0.266 \pm 0.014$ |
| softmax | $0.245 \pm 0.010$ |

## C.4  Memorization in SL vs. SSL

We conducted an additional experiment where we trained a ResNet9 on CIFAR100 with SSL (SimCLR) and SL (cross-entropy loss). For the SL model, we remove the classification layer to turn it into an encoder. Then, we report linear probing accuracies on multiple downstream tasks in Table 20. Our results highlight that the SL pretrained encoders exhibit a significantly higher downstream accuracy on their pretraining dataset than the SSL encoder. We assume that this is because of the class memorization. In contrast, the SL pretrained encoders perform significantly worse on other datasets than the SSL pretrained encoders since they might overfit the representations to their classes rather than provide more general (instance-based) representations as the SSL encoders. Additionally, we note that prior work has shown that the MAE encoder provides the highest performance when a few last layers are fine-tuned. The results in the original MAE paper in Figure 9 [30] indicate that fine-tuning a few last layers/blocks (e.g., 4 or 6 blocks out of 24 in ViT-Large) can achieve accuracy close to full fine-tuning (when all 24 blocks are fine-tuned). This is in line with our observation that the difference between UnitMem and ClassMem is the highest in the few last layers/blocks. Thus, fine-tuning only these last layers/blocks suffices for good downstream performance.

Table 18: **Layer-based Memorization Scores.** We present the layer-wise memorization of an SSL encoder pretrained on CIFAR10 using ResNet9 with SimCLR. The 1st column represents the IDs of convolutional layers and the 2nd column shows the name of the layers. Residual$N$ denotes that the residual connection comes from the previous $N$-th convolutional layer. We report `LayerMem` across the 100 randomly chosen training data points, their $\Delta$`LayerMem` (denoted as $\Delta$LM), followed by `LayerMem` for only the Top 50 memorized data points, their $\Delta$`LayerMem` (denoted as $\Delta$Top50), and `LayerMem` for only the Least 50 memorized data points. The projection *head* layer (denoted as head) is used only for training.

| ID | Name | LayerMem | $\Delta$LM | LayerMem Top50 | $\Delta$Top50 | LayerMem Least50 |
|----|------|----------|------------|----------------|---------------|------------------|
| 1 | Conv1 | 0.091 | - | 0.144 | - | 0.003 |
| - | BN1 | 0.091 | 0.000 | 0.144 | 0 | 0.004 |
| - | MaxPool | 0.097 | 0.006 | 0.158 | 0.014 | 0.004 |
| 2 | Conv2-0 | 0.123 | 0.026 | 0.225 | 0.067 | 0.012 |
| - | BN2-0 | 0.124 | 0.001 | 0.225 | 0 | 0.012 |
| - | MaxPool | 0.128 | 0.004 | 0.236 | 0.011 | 0.013 |
| 3 | Conv2-1 | 0.154 | 0.026 | 0.308 | 0.072 | 0.022 |
| 4 | Conv2-2 | 0.183 | 0.029 | 0.402 | 0.094 | 0.031 |
| - | Residual2 | 0.185 | 0.002 | 0.403 | 0.01 | 0.041 |
| 5 | Conv3 | 0.212 | 0.027 | 0.479 | 0.076 | 0.051 |
| - | BN3 | 0.211 | -0.001 | 0.480 | 0.001 | 0.051 |
| - | MaxPool | 0.215 | 0.004 | 0.486 | 0.006 | 0.050 |
| 6 | Conv4-0 | 0.246 | 0.031 | 0.599 | 0.113 | 0.061 |
| - | BN4-0 | 0.244 | -0.002 | 0.600 | 0.001 | 0.060 |
| - | MaxPool | 0.247 | 0.003 | 0.603 | 0.003 | 0.061 |
| 7 | Conv4-1 | 0.276 | 0.029 | 0.697 | 0.094 | 0.071 |
| 8 | Conv4-2 | 0.308 | 0.032 | 0.817 | 0.120 | 0.073 |
| - | Residual6 | 0.311 | 0.003 | 0.817 | 0 | 0.086 |
| - | head | 0.319 | 0.008 | 0.819 | 0.002 | 0.097 |
| - | MaxPool | 0.318 | -0.001 | 0.819 | 0 | 0.096 |
| - | FC | 0.192 | -0.126 | 0.409 | -0.410 | 0.071 |

Table 19: `LayerMem` **is not sensitive to the number of samples used for its calculation.** We pre-train a ResNet9 using SimCLR on CIFAR10 and determined `LayerMem` on batches of different sizes. For each batch size, we use three independent seeds (*i.e.,* different batch compositions) and report the average `LayerMem` score and its standard deviation. The results show that the reported `LayerMem` score is, indeed, similar across all setups. This indicates `LayerMem` 's insensitivity to the choice of the batch used to compute it.

| Layer | 100 samples | 500 samples | 1000 samples | 5000 samples |
|-------|-------------|-------------|--------------|--------------|
| 1 | 0.092 ± 8e-4 | 0.093 ± 7e-4 | 0.089 ± 7e-4 | 0.091 ± 8e-4 |
| 2 | 0.122 ± 9e-4 | 0.124 ± 1e-3 | 0.122 ± 8e-4 | 0.121 ± 7e-4 |
| 3 | 0.150 ± 1e-3 | 0.154 ± 9e-4 | 0.151 ± 8e-4 | 0.152 ± 6e-4 |
| 4 | 0.181 ± 1e-3 | 0.182 ± 8e-4 | 0.180 ± 8e-4 | 0.181 ± 8e-4 |
| Res2 | 0.184 ± 9e-4 | 0.185 ± 8e-4 | 0.183 ± 9e-4 | 0.184 ± 6e-4 |
| 5 | 0.213 ± 8e-4 | 0.213 ± 8e-4 | 0.212 ± 7e-4 | 0.212 ± 8e-4 |
| 6 | 0.246 ± 1e-3 | 0.249 ± 7e-4 | 0.247 ± 8e-4 | 0.245 ± 9e-4 |
| 7 | 0.277 ± 7e-4 | 0.281 ± 9e-4 | 0.277 ± 8e-4 | 0.276 ± 4e-4 |
| 8 | 0.309 ± 9e-4 | 0.314 ± 8e-4 | 0.310 ± 7e-4 | 0.307 ± 7e-4 |
| Res6 | 0.310 ± 1e-3 | 0.316 ± 1e-3 | 0.313 ± 8e-4 | 0.309 ± 9e-4 |

## C.5 Visualization for Variability and Consistency of Memorization cross Different Layers.

We present the top 10 most memorized samples of each layer for the ResNet9 vision encoder trained with the CIFAR10 dataset in Figure 13. The results show that the overlap within the top 10 most memorized samples between adjacent layers is usually high but decreases the further the layers are separated. This aligns with the results of overlap rate and Kendall's Tau test reported in Table 3.

## C.6 Layer-based Memorization Across Different SSL Frameworks and Datasets

We present the full results for the Table 4, which show that the layer-based memorization is similar across encoders trained with different SSL frameworks. The results for the ResNet50 architecture trained with SimCLR and DINO using the ImageNet dataset are presented in Table 29, and the results for the ViT-Base architecture trained with MAE and DINO using the ImageNet dataset are shown in Table 31.

Table 20: **Comparing the impact of memorization on downstream generalization between SSL and SL.** We train a ResNet9 on CIFAR100 with SSL (pretrained on CIFAR100 using SimCLR and SL (cross-entropy loss, trained until convergence). For the SL model, we remove the classification layer to turn it into an encoder. Then, we report linear probing accuracies on multiple downstream tasks in

| Encoder | CIFAR100 | CIFAR10 | STL10 | SVHN |
|---|---|---|---|---|
| SSL | 65.4% ± 0.98% | 57.6% ± 0.87% | 48.7% ± 0.98% | 59.2% ± 0.76% |
| SL (trained until convergence on CIFAR100, last layer removed) | 66.1% ± 1.12% | 56.7% ± 0.83% | 46.1% ± 1.04% | 58.6% ± 0.82% |

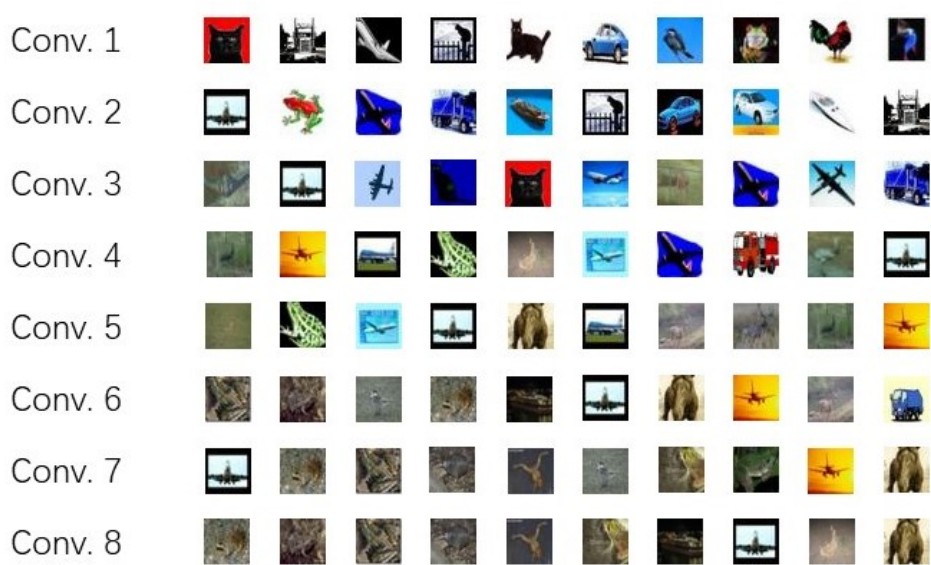

Figure 13: **The most memorized samples per layer according to** `LayerMem`.

## C.7 Verification of Layer-Based Memorization

To analyze whether our `LayerMem` metric and its $\Delta$ variant indeed localize memorization correctly, we first replace different layers of an encoder and then compute linear probing accuracy on various downstream tasks. Since prior work shows that memorization in SSL is required for downstream generalization [47], we expect the highest performance drop when replacing the layers identified as most memorizing. We verify this hypothesis and train a ResNet9 encoder $f_1$ on the CIFAR10 dataset and compute the `LayerMem` and $\Delta$`LayerMem` scores per layer. Then, we select the three most memorized, random, and least memorized layers and replace them with the corresponding layers from a ResNet9 trained on STL10 ($f_2$). Our results in Table 21 show that the highest linear probing accuracy drop on the CIFAR10 test set for $f_1$ is caused by replacing the three most memorized layers from $f_1$ according to the $\Delta$`LayerMem` score. The second biggest drop is observed when replacing according to `LayerMem`, highlighting that indeed our `LayerMem` metric and its $\Delta$ variant identify the most crucial layers in SSL encoders for memorization. Surprisingly, the replacement of the layers in $f_1$ with the corresponding layers from $f_2$ causes the biggest simultaneous increase in the downstream accuracy for the STL10 dataset. We observe the same trends when $f_2$ is trained on SVHN (Table 22b), for replacing single layers in ResNet9 (Appendix C.7), and replacing whole blocks in ResNet50 (Appendix C.7) instead of only individual layers as we present in Appendix C.7. The above analysis verifies that the `LayerMem` score and its $\Delta$ variant identify the most crucial layers in SSL encoders. They further strengthen the claims that memorization is required for generalization [23, 24, 47].

**Replacing layers in ResNet9 for SVHN.** In Table 22b, we show the effect of replacing the most and least vs random layers of a CIFAR10 trained ResNet9 on the downstream performance. We replace the layers with the corresponding ones from a ResNet9 encoder trained on SVHN.

Table 21: **Replacing the most/least memorized layers according to $\Delta$LayerMem causes the most/least changes in downstream performance.** We study the effect of replacing layers of the ResNet9 encoder trained on CIFAR10 with layers from another ResNet9 encoder trained on STL10 and report the linear probing accuracy on the CIFAR10 and STL10 test sets. Results for the impact of replacing any combination of 1, 2, and 3 layers on downstream accuracy are shown in Appendix C.12.

| Replacement Criteria | Replaced Layer(s) | CIFAR10 | STL10 |
|---|---|---|---|
| *None (Baseline)* | *None* | 69.08% $\pm$ 1.05% | 17.81% $\pm$ 0.92% |
| **Most Memorized LayerMem** | **4 6 8** | **36.59% $\pm$ 1.13%** | **32.33% $\pm$ 0.88%** |
| Most Memorized LayerMem | 6 7 8 | 39.07% $\pm$ 1.05% | 29.82% $\pm$ 0.91% |
| Random | 4 5 7 | 43.22% $\pm$ 1.08% | 25.89% $\pm$ 0.93% |
| Least Memorized LayerMem | 2 3 4 | 49.95% $\pm$ 1.21% | 24.71% $\pm$ 0.99% |
| **Least Memorized $\Delta$LayerMem** | **2 3 5** | **59.14% $\pm$ 0.91%** | **23.10% $\pm$ 1.06%** |

Table 22: **Evaluating the effect of replacing layers of the ResNet9 encoder pre-trained on CIFAR10 with layers from ResNet9 pre-trained on STL10.** We report the linear probing accuracy of ResNet9 with the replaced layers and tested on the CIFAR10, STL10 test sets.

(a) CIFAR10 & STL10

| Replaced Criterium | Replaced Layer(s) | CIFAR10 | STL10 |
|---|---|---|---|
| *None (Baseline)* | *None* | 69.08% $\pm$ 1.05% | 17.81% $\pm$ 0.92% |
| Most Memorized (delta) | 4 6 8 | 36.59% $\pm$ 1.13% | 32.33% $\pm$ 0.88% |
| Most Memorized (absolute) | 6 7 8 | 39.07% $\pm$ 1.05% | 29.82% $\pm$ 0.91% |
| Random | 4 5 7 | 43.22% $\pm$ 1.08% | 25.89% $\pm$ 0.93% |
| Least Memorized (delta) | 2 3 5 | 59.14% $\pm$ 0.91% | 23.10% $\pm$ 1.06% |
| Least Memorized (absolute) | 2 3 4 | 49.95% $\pm$ 1.21% | 24.71% $\pm$ 0.99% |

(b) CIFAR10 & SVHN

| Replaced Criterium | Replaced Layer(s) | CIFAR10 | SVHN |
|---|---|---|---|
| *None (Baseline)* | *None* | 69.08% $\pm$ 1.05% | 19.33% $\pm$ 0.65% |
| Most Memorized (delta) | 4 6 8 | 33.07% $\pm$ 1.51% | 34.05% $\pm$ 1.01% |
| Most Memorized (absolute) | 6 7 8 | 34.97% $\pm$ 0.84% | 31.87% $\pm$ 1.21% |
| Random | 4 5 7 | 39.28% $\pm$ 0.74% | 26.04% $\pm$ 0.82% |
| Least Memorized (delta) | 2 3 5 | 52.81% $\pm$ 1.03% | 21.05% $\pm$ 0.89% |
| Least Memorized (absolute) | 2 3 4 | 45.39% $\pm$ 1.10% | 24.66% $\pm$ 0.57% |

**Replacing blocks in ResNet50 trained on CIFAR10 with SimCLR.** We present the results for replacing blocks in ResNet50 trained on CIFAR10 using SimCLR in Appendix C.7.

**Statistics of Batch-Norm layer for different datasets.** Batch-norm layers between different datasets might have different statistics. This could impact the downstream performance. To investigate the changes, we measured the cosine similarity between the weights and biases of the batch-norm layers for two encoders (trained on CIFAR10 and STL10, respectively). The results in Table 25 show a high per-layer cosine similarity (average over all layers=0.823). This suggests that the statistics are similar, hence, no adjustment is required. We hypothesize that the similarity stems from the fact that the data distributions are similar and that we normalize both input datasets according to the ImageNet normalization parameters.

## C.8 LayerMem **with Different Distance Metrics**

In addition to the $\ell_2$ distance, we also used 3 other distance metrics ($\ell_1$, cosine similarity, and angular distance) to evaluate the stability of LayerMem. Our results in Table 26 highlight that **1)** the memorization scores are very similar, independent of the choice of the distance metric, and **2)** the most memorizing layers according to and $\Delta$LayerMem are the same over all metrics. This suggests that our findings are independent of the choice of distance metric.

Table 23: **Evaluating the effect of replacing layers of the ResNet9 encoder pre-trained on CIFAR10 with layers from ResNet9 pre-trained on STL10**. We report the linear probing accuracy of ResNet9 with the replaced layers and tested on the CIFAR10, STL10 test sets.

(a) CIFAR10 & STL10

| Replaced Criterium | Replaced Layer(s) | CIFAR10 | STL10 |
|---|---|---|---|
| *None (Baseline)* | *None* | 69.08% ± 1.05% | 17.81% ± 0.92% |
| Most Memorized (delta) | 6 | 59.84% ± 1.20% | 21.98% ± 0.41% |
| Most Memorized (absolute) | 8 | 60.02% ± 0.94% | 21.67% ± 0.72% |
| Random | 5 | 62.98% ± 0.57% | 20.44% ± 0.85% |
| Least Memorized (delta) | 2 | 65.52% ± 0.74% | 18.94% ± 0.63% |
| Least Memorized (absolute) | 3 | 64.21% ± 1.08% | 18.89% ± 0.81% |

(b) CIFAR10 & SVHN

| Replaced Criterium | Replaced Layer(s) | CIFAR10 | SVHN |
|---|---|---|---|
| None | | 69.08% ± 1.05% | 19.33% ± 0.65% |
| Most Memorized (delta) | 6 | 59.22% ± 0.97% | 22.47% ± 0.57% |
| Most Memorized (absolute) | 8 | 59.69% ± 1.04% | 21.60% ± 0.92% |
| Random | 5 | 61.07% ± 1.12% | 21.09% ± 0.69% |
| Least Memorized (delta) | 2 | 62.93% ± 1.08% | 20.44% ± 0.71% |
| Least Memorized (absolute) | 3 | 62.35% ± 0.81% | 20.18% ± 0.98% |

Table 24: **Evaluating the effect of replacing blocks of ResNet50 pre-trained on CIFAR10 with blocks from ResNet50 pre-trained on STL10 and SVHN**. The accuracy in the table is the linear probing accuracy of ResNet50 on CIFAR10. We replace block 3 of conv layers, which was selected according to the biggest $\Delta$ LayerMem between two layers (not the absolute value of the LayerMem score of the layers).

(a) CIFAR10 & STL10

| Replaced Criterium | Replaced Layer(s) | CIFAR10 | STL10 |
|---|---|---|---|
| None | / | 77.12% ± 1.42% | 18.22% ± 0.88% |
| Most Memorized | C4_B6 C3_B4 C2_B3 | 43.66% ± 1.20% | 25.78% ± 0.95% |
| Random | C3_B2 C4_B4 C5_B2 | 51.09% ± 1.01% | 22.55% ± 1.17% |
| Least Memorized | C2_B1 C2_B2 C3_B3 | 57.41% ± 0.74% | 20.10% ± 1.11% |

(b) CIFAR10 & SVHN

| Replaced Criterium | Replaced Layer(s) | CIFAR10 | SVHN |
|---|---|---|---|
| None | / | 77.12% ± 1.42% | 28.44% ± 1.23% |
| Most Memorized | C4_B6 C3_B4 C2_B3 | 35.21% ± 0.94% | 38.11% ± 1.08% |
| Random | C3_B2 C4_B4 C5_B2 | 44.19% ± 0.97% | 32.44% ± 1.25% |
| Least Memorized | C2_B1 C2_B2 C3_B3 | 49.06% ± 1.31% | 29.98% ± 0.85% |

## C.9 LayerMem **with Different Augmentation Strength**

The results, reported in Table 27 highlight that stronger training augmentations reduce LayerMem.

## C.10 LayerMem **with Different Initialization of Trainable parameters**

We performed an additional experiment where we trained encoders f and g independently with a different random seed (yielding f' and g') to study how random initialization of trainable parameters can affect the memorization of final vision encoder.The results are reported in Table 28. We compared the overlap in most memorized samples between encoder f (from the paper) and f'. The results (Table 4, attached PDF) show that overlap is overall high (min. 69% in layer 2) and increases in the later layers (max. 90%, final layer).

Table 25: **Cosine similarities between batch-norm layer outputs for ResNet9 trained on CIFAR10 and STL10.** We normalize the training data according to the ImageNet parameters and train the encoders using SimCLR. We calculate the cosine similarity over the weights ($\gamma$) and the bias ($\beta$) of the respective encoders' trained batch-norm layers.

| Layer | 1 | 2 | 3 | 4 | 5 | 6 | 7 | 8 |
|---|---|---|---|---|---|---|---|---|
| Cosine Similarity | 0.797 | 0.844 | 0.823 | 0.811 | 0.779 | 0.805 | 0.847 | 0.881 |

Table 26: `LayerMem` **(LM) and** $\Delta$**-LM under different distance metrics.** We report for $\ell_1$, $\ell_2$ (see original submission), cosine similarity (Cos. Sim), and angular distance (Ang. Dist). The results highlight that our memorization measure is independent of the underlying metric. (ResNet9, CIFAR10, SimCLR).

| Layer | $\ell_1$ LM | $\ell_1$ $\Delta$LM | $\ell_2$ LM | $\ell_2$ $\Delta$LM | Cos. Sim. LM | Cos. Sim. $\Delta$LM | Ang. Dist. LM | Ang. Dist. $\Delta$LM |
|---|---|---|---|---|---|---|---|---|
| 1 | 0.099 | - | 0.091 | - | 0.104 | - | 0.096 | - |
| 2 | 0.128 | 0.029 | 0.123 | 0.032 | 0.134 | 0.030 | 0.128 | 0.032 |
| 3 | 0.159 | 0.031 | 0.154 | 0.031 | 0.163 | 0.029 | 0.160 | 0.032 |
| 4 | 0.187 | 0.028 | 0.183 | 0.029 | 0.190 | 0.027 | 0.191 | 0.031 |
| Res2 | 0.192 | 0.005 | 0.185 | 0.002 | 0.193 | 0.003 | 0.193 | 0.002 |
| 5 | 0.221 | 0.029 | 0.212 | 0.027 | 0.220 | 0.027 | 0.222 | 0.029 |
| 6 | 0.256 | 0.035 | 0.246 | 0.034 | 0.256 | 0.036 | 0.259 | 0.037 |
| 7 | 0.289 | 0.033 | 0.276 | 0.030 | 0.288 | 0.032 | 0.293 | 0.034 |
| 8 | 0.325 | 0.036 | 0.308 | 0.032 | 0.321 | 0.033 | 0.328 | 0.035 |
| Res6 | 0.329 | 0.004 | 0.311 | 0.003 | 0.323 | 0.002 | 0.329 | 0.001 |

## C.11 Impact of Layer Replacement on Layer Memorization

According to the definition of SSLMem Equation (2), we let the representations of a given input data point $x$ pass through the same (replaced) layer in both f and g. We show the `LayerMem` and $\Delta$LayerMem scores after replacing a single layer in the ResNet9 encoder pre-trained using SimCLR on the CIFAR10 dataset in Table 23 and Table 30. The `LayerMem` score of the replaced layer always drops as expected since this layer does not memorize any original training data points. The decrease in LayerMem between the initial and replaced layers is smaller in the earlier layers (e.g., 1st layer) as compared to the later layers (e.g., 6th layer). This might be because, in general, the earlier layers from different models might be more similar as they are responsible for extracting general features instead of specific ones for a given dataset. The most important take-away from these experiments is that the $\Delta$LayerMem is not affected significantly and its values show the same trends after the layer replacement.

## C.12 Layer Replacement for Single, Two, and Three Layers at a Time

We perform the experiment with the replacement of 1 layer in Table 30, 2 layers Table 32, and 3 layers Table 33. The following results confirm our results from Table 21 in the main paper. When only a single layer is replaced, then the 6th (not the last layer) is the most important one. This layer had the highest `LayerMem` score. Note that the replacement of the 6th layer causes the highest drop in accuracy on the original CIFAR10 dataset and the highest gain in accuracy on STL10. Next, when two layers are replaced, then layers 6th and 8th play the most important roles, where

Table 27: `LayerMem` (LM) **and** $\Delta$**-LM under different augmentation sets used during training.** We use the augmentations defined in Appendix B during training and metric calculation. The results show that stronger augmentations reduce memorization. (ResNet9, CIFAR10, SimCLR).

| Layer | weak | | normal | | strong | |
|---|---|---|---|---|---|---|
| | LM | $\Delta$LM | LM | $\Delta$LM | LM | $\Delta$LM |
| 1 | 0.092 | - | 0.091 | - | 0.089 | - |
| 2 | 0.123 | 0.031 | 0.123 | 0.032 | 0.120 | 0.031 |
| 3 | 0.154 | 0.031 | 0.154 | 0.031 | 0.150 | 0.030 |
| 4 | 0.184 | 0.030 | 0.183 | 0.029 | 0.178 | 0.028 |
| Res2 | 0.187 | 0.003 | 0.185 | 0.002 | 0.181 | 0.003 |
| 5 | 0.215 | 0.028 | 0.212 | 0.027 | 0.208 | 0.027 |
| 6 | 0.249 | 0.034 | 0.246 | 0.034 | 0.241 | 0.033 |
| 7 | 0.280 | 0.031 | 0.276 | 0.030 | 0.269 | 0.028 |
| 8 | 0.313 | 0.033 | 0.308 | 0.032 | 0.300 | 0.031 |
| Res6 | 0.315 | 0.002 | 0.311 | 0.003 | 0.302 | 0.002 |

Table 28: **Overlap in 100 most memorized samples according to** `LayerMem` **between 2 different encoders.** We train encoders with different seeds and report the per-layer overlap in their most memorized samples. We observe an overall high overlap, especially in the last layer.

| Layer | 1 | 2 | 3 | 4 | 5 | 6 | 7 | 8 |
|---|---|---|---|---|---|---|---|---|
| Overlap % | 73 | 69 | 75 | 89 | 85 | 88 | 86 | 90 |

their replacement with layers from the encoder trained on STL10 causes the highest drop on CIFAR10 and the highest performance increase on STL10. This is contrary to the common intuition, which would suggest the replacement of the last two layers instead.

## D   Additional Setup

### D.1   Class Selectivity

We denote the class selectivity metric as `ClassSelectivity`. It was proposed by [39] to quantify a unit's discriminability between different classes and described more in the main part of the paper in Section 5. We derive the basic metric in more detail here.

To compute the `ClassSelectivity` metric per unit $u$, first the class-conditional mean activity is calculated for the test dataset $\bar{\mathcal{D}}$. We denote each test data point as $\bar{x}_i \in \bar{\mathcal{D}}$. We assume $M$ classes $C_{j=1}^M$, each with its corresponding test data points $\bar{x}_c \in C_j$, where $c = 1, 2, ..., |C_j|$.

We define the mean activation $\bar{\mu}$ of unit $u$ for class $C_j$ as

$$\bar{\mu}_u(C_j) = \frac{1}{|C_j|} \sum_{\bar{x}_c \in C_j} \text{activation}_u(\bar{x}_c), \tag{9}$$

where the activation for convolutional feature maps is averaged across all elements of the feature map. Further, for the unit $u$, we compute the maximum mean activation $\bar{\mu}_{max,u}$ across all classes $C$, where $M = |C|$, as

$$\bar{\mu}_{max,u} = \max(\{\bar{\mu}_u(\bar{x}_i)\}_{i=1}^M). \tag{10}$$

Let $p$ be the index location of the maximum mean activation $\bar{\mu}_u(\bar{x}_p)$, *i.e.,* the $argmax$. Then, we calculate the corresponding mean activity $\bar{\mu}_{-max,u}$ across all the remaining $M - 1$ classes as

$$\bar{\mu}_{-max,u} = \frac{1}{M - 1} \sum_{j=1, j \neq p}^{M} \bar{\mu}_u(C_j). \tag{11}$$

Finally, the class selectivity is then calculated as follows

$$\texttt{ClassSelectivity}(u) = \frac{\bar{\mu}_{max,u} - \bar{\mu}_{-max,u}}{\bar{\mu}_{max,u} + \bar{\mu}_{-max,u}}, \tag{12}$$

where $\bar{\mu}_{max,u}$ represents the highest class-conditional mean activity and $\bar{\mu}_{-max,u}$ denotes the mean activity across all other classes (for unit $u$ and computed on the test dataset $\bar{\mathcal{D}}$).

### D.2 Class Memorization

We use a similar definition as $\texttt{ClassSelectivity}$ for $\texttt{ClassMem}$, which measures how much a given unit is responsible for the memorization of a class. While $\texttt{ClassSelectivity}$ is calculated on the *test set*, we compute $\texttt{ClassMem}$ on the *training dataset*.

To compute the $\texttt{ClassMem}$ metric per unit $u$, first, the class-conditional mean activity is calculated for the training dataset $\mathcal{D}'$. We denote each train data point as $x_i \in \mathcal{D}'$. We assume $M$ classes $C_{j=1}^M$, each with its corresponding train data points $x_c \in C_j$, where $c = 1, 2, ..., |C_j|$.

We define the mean activation $\tilde{\mu}$ of unit $u$ for class $C_j$ as

$$\tilde{\mu}_u(C_j) = \frac{1}{|C_j|} \sum_{x_c \in C_j} \text{activation}_u(x_c), \tag{13}$$

where the activation for convolutional feature maps is averaged across all elements of the feature map. Further, for the unit $u$, we compute the maximum mean activation $\tilde{\mu}_{max,u}$ across all classes $C$, where $M = |C|$, as

$$\tilde{\mu}_{max,u} = \max(\{\tilde{\mu}_u(x_i)\}_{i=1}^M). \tag{14}$$

Let $p$ be the index location of the maximum mean activation $\tilde{\mu}_u(\tilde{x}_p)$, *i.e.,* the $argmax$. Then, we calculate the corresponding mean activity $\tilde{\mu}_{-max,u}$ across all the remaining $M - 1$ classes as

$$\tilde{\mu}_{-max,u} = \frac{1}{M - 1} \sum_{j=1, j \neq p}^{M} \tilde{\mu}_u(C_j). \tag{15}$$

Finally, the class Memorization is then calculated as follows

$$\texttt{ClassMem}(u) = \frac{\tilde{\mu}_{max,u} - \tilde{\mu}_{-max,u}}{\tilde{\mu}_{max,u} + \tilde{\mu}_{-max,u}}, \tag{16}$$

where $\tilde{\mu}_{max,u}$ represents the highest class-conditional mean activity and $\tilde{\mu}_{-max,u}$ denotes the mean activity across all other classes (for unit $u$ and computed on the train dataset $\mathcal{D}'$).

# E   Extended Related Work

**Localizing Memorization on the Level of Individual Units.**    In Section 5, we considered memorization from the perspective of individual units and identified that pruning the least/most memorized units according to `UnitMem` preserves the least/most performance (as shown in Table Table 6). The work by Maini et al. [35] characterized individual examples as mislabeled based on the low number of channels or filters that need to be zeroed out to flip the prediction. They observe that significantly more neurons need to be zeroed out to flip clean examples compared to mislabeled ones.

A similar experiment in the SSL domain could potentially reveal a similar trend, where noisy examples are harder to learn and primarily influence a small number of units. However, SSL encoders do not have discrete output changes from zeroing out individual units. One could pre-train the encoder and add linear probing, but this would require labels for the SSL training set, making it inapplicable. Even with labeled data and fine-tuning, identifying noisy SSL examples based on the `SSLMem` score may not match mislabeled examples in SL. The lack of a discrete oracle and the potential mismatch between noisy SSL and mislabeled SL examples makes it difficult to identify individual units responsible for predictions of selected examples using prior methods.

# F   Impact & Limitations

The fact that memorization can enable privacy attacks, such as data extraction [9, 10, 12], has been established in prior work. Yet, this paper advances the field of machine learning towards a novel fundamental understanding on where in SSL encoders memorization happens, and how memorization differs between standard SL models and SSL encoders. Our insights hold the potential to yield societal benefits in the form of the design of novel methods to reduce memorization, improve fine-tuning, and yield better model pruning algorithms.

## G  Additional Results

Table 29: **All-layer memorization.** We train the ResNet50 encoder using SimCLR and DINO SSL frameworks on the ImageNet dataset. We report the full results with the LayerMem and $\Delta$LayerMem scores for each layer.

| ResNet50 Layer | SimCLR | | DINO | |
| --- | --- | --- | --- | --- |
| | LayerMem | $\Delta$LayerMem | LayerMem | $\Delta$LayerMem |
| conv1 | $0.038 \pm 0.001$ | - | $0.040 \pm 0.002$ | - |
| max pool | $0.039 \pm 0.002$ | 0.001 | $0.040 \pm 0.002$ | 0.000 |
| conv2-1 | $0.041 \pm 0.002$ | 0.002 | $0.043 \pm 0.001$ | 0.003 |
| conv2-2 | $0.044 \pm 0.002$ | 0.003 | $0.045 \pm 0.001$ | 0.002 |
| conv2-3 | $0.048 \pm 0.001$ | 0.004 | $0.048 \pm 0.002$ | 0.003 |
| conv2-4 | $0.052 \pm 0.002$ | 0.004 | $0.052 \pm 0.001$ | 0.004 |
| conv2-5 | $0.055 \pm 0.001$ | 0.003 | $0.056 \pm 0.001$ | 0.004 |
| conv2-6 | $0.059 \pm 0.002$ | 0.004 | $0.060 \pm 0.001$ | 0.004 |
| conv2-7 | $0.063 \pm 0.001$ | 0.004 | $0.065 \pm 0.001$ | 0.005 |
| conv2-8 | $0.068 \pm 0.001$ | 0.005 | $0.069 \pm 0.002$ | 0.004 |
| conv2-9 | $0.072 \pm 0.002$ | 0.004 | $0.073 \pm 0.001$ | 0.004 |
| conv3-1 | $0.077 \pm 0.002$ | 0.005 | $0.078 \pm 0.002$ | 0.005 |
| conv3-2 | $0.081 \pm 0.003$ | 0.004 | $0.083 \pm 0.001$ | 0.005 |
| conv3-3 | $0.086 \pm 0.002$ | 0.005 | $0.088 \pm 0.002$ | 0.005 |
| conv3-4 | $0.092 \pm 0.001$ | 0.006 | $0.094 \pm 0.001$ | 0.006 |
| conv3-5 | $0.097 \pm 0.001$ | 0.005 | $0.099 \pm 0.002$ | 0.005 |
| conv3-6 | $0.103 \pm 0.002$ | 0.006 | $0.104 \pm 0.001$ | 0.005 |
| conv3-7 | $0.108 \pm 0.002$ | 0.005 | $0.110 \pm 0.001$ | 0.006 |
| conv3-8 | $0.112 \pm 0.002$ | 0.004 | $0.115 \pm 0.002$ | 0.005 |
| conv3-9 | $0.117 \pm 0.001$ | 0.005 | $0.120 \pm 0.001$ | 0.005 |
| conv3-10 | $0.123 \pm 0.002$ | 0.006 | $0.126 \pm 0.002$ | 0.006 |
| conv3-11 | $0.128 \pm 0.001$ | 0.005 | $0.131 \pm 0.003$ | 0.005 |
| conv3-12 | $0.134 \pm 0.002$ | 0.006 | $0.136 \pm 0.002$ | 0.005 |
| conv4-1 | $0.139 \pm 0.002$ | 0.005 | $0.142 \pm 0.002$ | 0.006 |
| conv4-2 | $0.145 \pm 0.002$ | 0.006 | $0.148 \pm 0.003$ | 0.006 |
| conv4-3 | $0.150 \pm 0.003$ | 0.005 | $0.153 \pm 0.002$ | 0.005 |
| conv4-4 | $0.156 \pm 0.003$ | 0.006 | $0.159 \pm 0.003$ | 0.006 |
| conv4-5 | $0.161 \pm 0.002$ | 0.005 | $0.164 \pm 0.003$ | 0.005 |
| conv4-6 | $0.166 \pm 0.003$ | 0.005 | $0.169 \pm 0.004$ | 0.005 |
| conv4-7 | $0.172 \pm 0.004$ | 0.006 | $0.175 \pm 0.002$ | 0.006 |
| conv4-8 | $0.178 \pm 0.003$ | 0.006 | $0.181 \pm 0.003$ | 0.006 |
| conv4-9 | $0.183 \pm 0.002$ | 0.005 | $0.186 \pm 0.002$ | 0.005 |
| conv4-10 | $0.189 \pm 0.003$ | 0.006 | $0.192 \pm 0.003$ | 0.006 |
| conv4-11 | $0.194 \pm 0.004$ | 0.005 | $0.198 \pm 0.004$ | 0.006 |
| conv4-12 | $0.200 \pm 0.003$ | 0.006 | $0.203 \pm 0.005$ | 0.005 |
| conv4-13 | $0.207 \pm 0.006$ | 0.007 | $0.210 \pm 0.003$ | 0.007 |
| conv4-14 | $0.212 \pm 0.002$ | 0.005 | $0.216 \pm 0.004$ | 0.006 |
| conv4-15 | $0.218 \pm 0.003$ | 0.006 | $0.221 \pm 0.004$ | 0.005 |
| conv4-16 | $0.224 \pm 0.004$ | 0.006 | $0.228 \pm 0.005$ | 0.007 |
| conv4-17 | $0.229 \pm 0.003$ | 0.005 | $0.234 \pm 0.003$ | 0.006 |
| conv4-18 | $0.235 \pm 0.005$ | 0.006 | $0.240 \pm 0.003$ | 0.006 |
| conv5-1 | $0.242 \pm 0.003$ | 0.007 | $0.247 \pm 0.004$ | 0.007 |
| conv5-2 | $0.249 \pm 0.003$ | 0.007 | $0.254 \pm 0.005$ | 0.007 |
| conv5-3 | $0.257 \pm 0.002$ | 0.008 | $0.261 \pm 0.004$ | 0.007 |
| conv5-4 | $0.265 \pm 0.002$ | 0.008 | $0.269 \pm 0.005$ | 0.008 |
| conv5-5 | $0.272 \pm 0.004$ | 0.007 | $0.276 \pm 0.004$ | 0.007 |
| conv5-6 | $0.279 \pm 0.003$ | 0.007 | $0.283 \pm 0.003$ | 0.007 |
| conv5-7 | $0.287 \pm 0.003$ | 0.008 | $0.292 \pm 0.006$ | 0.009 |
| conv5-8 | $0.295 \pm 0.003$ | 0.008 | $0.300 \pm 0.004$ | 0.008 |
| conv5-9 | $0.304 \pm 0.005$ | 0.009 | $0.309 \pm 0.004$ | 0.009 |

Table 30: **Replace a single layer.** We follow the settings from the Table 21 (same encoder) and replace a single layer at a time.

| Replaced Layers | CIFAR10 | STL10 |
| --- | --- | --- |
| None | 69.37%±1.07% | 20.09%±0.64% |
| 1 | 63.07%±0.93% | 18.79%±0.62% |
| 2 | 65.19%±0.87% | 19.01%±0.77% |
| 3 | 64.47%±1.15% | 18.99%±0.81% |
| 4 | 60.29%±0.74% | 20.44%±0.66% |
| 5 | 62.74%±0.82% | 19.93%±0.59% |
| 6 | 59.91%±1.09% | 21.92%±0.67% |
| 7 | 60.77%±0.75% | 20.97%±0.52% |
| 8 | 60.04%±0.90% | 21.71%±0.58% |
| max(=layerX) | 65.19%±0.87%(2) | 21.92%±0.67%(6) |
| min(=layerX) | 59.91%±1.09%(6) | 18.79%±0.62%(2) |

Table 31: **All-layer memorization.** We train the ViT-Base encoder using MAE and DINO SSL frameworks on the ImageNet dataset. We report the full results with the LayerMem and $\Delta$LayerMem scores for each block.

| ViT-Base Block Number | MAE | | DINO | |
|---|---|---|---|---|
| | LayerMem | $\Delta$LayerMem | LayerMem | $\Delta$LayerMem |
| 1 | 0.019±0.001 | - | 0.019±0.001 | - |
| 2 | 0.037±0.001 | 0.011 | 0.036±0.002 | 0.012 |
| 3 | 0.055±0.002 | 0.013 | 0.056±0.003 | 0.012 |
| 4 | 0.075±0.002 | 0.013 | 0.077±0.002 | 0.014 |
| 5 | 0.095±0.002 | 0.012 | 0.096±0.004 | 0.013 |
| 6 | 0.118±0.004 | 0.016 | 0.119±0.006 | 0.015 |
| 7 | 0.139±0.003 | 0.015 | 0.142±0.004 | 0.014 |
| 8 | 0.163±0.005 | 0.018 | 0.168±0.005 | 0.017 |
| 9 | 0.188±0.004 | 0.017 | 0.193±0.003 | 0.018 |
| 10 | 0.215±0.006 | 0.018 | 0.219±0.006 | 0.019 |
| 11 | 0.243±0.005 | 0.021 | 0.247±0.005 | 0.020 |
| 12 | 0.271±0.003 | 0.020 | 0.275±0.004 | 0.019 |

Table 32: **Replace two layers.** We follow the setting from the Table 21 (same encoder) and replace two layers at a time.

| Replaced Layers | CIFAR10 | STL10 |
|---|---|---|
| None | 69.37% ± 1.07% | 18.44% ± 0.64% |
| 1 2 | 53.44% ± 0.90% | 20.31% ± 0.51% |
| 1 3 | 52.51% ± 0.83% | 20.80% ± 0.60% |
| 1 4 | 50.77% ± 0.78% | 22.16% ± 0.66% |
| 1 5 | 50.98% ± 0.97% | 22.02% ± 0.59% |
| 1 6 | 46.09% ± 0.77% | 21.46% ± 0.69% |
| 1 7 | 49.59% ± 0.89% | 24.87% ± 0.66% |
| 1 8 | 45.96% ± 0.94% | 21.66% ± 0.71% |
| 2 3 | 55.44% ± 0.73% | 19.31% ± 0.72% |
| 2 4 | 53.61% ± 0.92% | 24.18% ± 0.68% |
| 2 5 | 53.34% ± 1.06% | 20.39% ± 0.57% |
| 2 6 | 48.59% ± 0.81% | 23.32% ± 0.70% |
| 2 7 | 51.07% ± 1.13% | 21.97% ± 0.52% |
| 2 8 | 50.15% ± 0.82% | 22.57% ± 0.61% |
| 3 4 | 52.99% ± 1.01% | 21.09% ± 0.73% |
| 3 5 | 52.67% ± 0.90% | 21.00% ± 0.81% |
| 3 6 | 48.22% ± 0.79% | 23.48% ± 0.62% |
| 3 7 | 50.81% ± 0.86% | 22.09% ± 0.70% |
| 3 8 | 49.07% ± 0.92% | 23.19% ± 0.67% |
| 4 5 | 50.49% ± 0.96% | 22.41% ± 0.55% |
| 4 6 | 44.88% ± 0.91% | 23.61% ± 0.49% |
| 4 7 | 46.38% ± 1.13% | 24.04% ± 0.62% |
| 4 8 | 45.09% ± 0.75% | 24.11% ± 0.66% |
| 5 6 | 46.02% ± 1.07% | 24.29% ± 0.81% |
| 5 7 | 49.21% ± 1.00% | 22.99% ± 0.80% |
| 5 8 | 45.71% ± 0.94% | 24.33% ± 0.73% |
| 6 7 | 44.76% ± 0.88% | 24.90% ± 0.54% |
| 6 8 | 44.13% ± 1.01% | 25.08% ± 0.64% |
| 7 8 | 44.98% ± 0.94% | 24.91% ± 0.81% |
| max (=layerX) | 55.44% ± 0.73% (2 3) | 25.08% ± 0.64% (6 8) |
| min (=layerX) | 44.13% ± 1.01% (6 8) | 19.31% ± 0.72% (2 3) |

Table 33: **Replace three layers.** We follow the settings from the Table 21 (same encoder) and replace three layers at a time.

| Replaced Layers | CIFAR10 | STL10 |
|---|---|---|
| None | 69.37% ± 1.07% | 18.44% ± 0.64% |
| 1 2 3 | 49.77% ± 0.66% | 21.95% ± 0.42% |
| 1 2 4 | 48.33% ± 0.65% | 23.28% ± 0.77% |
| 1 2 5 | 49.51% ± 0.70% | 22.18% ± 0.57% |
| 1 2 6 | 45.31% ± 0.59% | 26.41% ± 0.53% |
| 1 2 7 | 48.99% ± 0.84% | 22.54% ± 0.57% |
| 1 2 8 | 46.29% ± 0.48% | 25.74% ± 0.62% |
| 1 3 4 | 47.66% ± 0.52% | 24.37% ± 0.42% |
| 1 3 5 | 49.04% ± 0.65% | 22.20% ± 0.66% |
| 1 3 6 | 45.19% ± 0.72% | 26.57% ± 0.61% |
| 1 3 7 | 48.19% ± 0.58% | 23.76% ± 0.82% |
| 1 3 8 | 46.20% ± 0.83% | 25.67% ± 0.60% |
| 1 4 5 | 47.35% ± 0.60% | 24.99% ± 0.58% |
| 1 4 6 | 43.89% ± 0.70% | 29.57% ± 0.67% |
| 1 4 7 | 44.53% ± 0.66% | 27.55% ± 0.71% |
| 1 4 8 | 43.94% ± 0.78% | 29.26% ± 0.52% |
| 1 5 6 | 44.23% ± 0.70% | 27.89% ± 0.52% |
| 1 5 7 | 48.03% ± 0.55% | 23.30% ± 0.63% |
| 1 5 8 | 44.71% ± 0.71% | 26.99% ± 0.58% |
| 1 6 7 | 43.30% ± 0.44% | 29.81% ± 0.57% |
| 1 6 8 | 41.72% ± 0.70% | 30.71% ± 0.67% |
| 1 7 8 | 44.59% ± 0.83% | 28.06% ± 0.47% |
| 2 3 4 | 48.89% ± 0.38% | 22.76% ± 0.38% |
| 2 3 5 | 50.48% ± 0.67% | 21.71% ± 0.60% |
| 2 3 6 | 46.98% ± 0.57% | 25.68% ± 0.46% |
| 2 3 7 | 49.81% ± 0.62% | 22.31% ± 0.49% |
| 2 3 8 | 48.07% ± 0.93% | 23.74% ± 0.70% |
| 2 4 5 | 48.55% ± 0.79% | 23.90% ± 0.82% |
| 2 4 6 | 44.99% ± 0.58% | 27.88% ± 0.57% |
| 2 4 7 | 47.78% ± 0.68% | 24.87% ± 0.75% |
| 2 4 8 | 45.41% ± 0.86% | 26.98% ± 0.51% |
| 2 5 6 | 45.91% ± 0.44% | 26.47% ± 0.60% |
| 2 5 7 | 48.37% ± 0.55% | 22.90% ± 0.50% |
| 2 5 8 | 47.18% ± 0.52% | 25.57% ± 0.81% |
| 2 6 7 | 45.62% ± 0.69% | 26.78% ± 0.48% |
| 2 6 8 | 42.99% ± 0.63% | 29.41% ± 0.43% |
| 2 7 8 | 46.89% ± 0.93% | 25.77% ± 0.63% |
| 3 4 5 | 47.90% ± 0.56% | 24.74% ± 0.48% |
| 3 4 6 | 43.32% ± 0.58% | 28.73% ± 0.60% |
| 3 4 7 | 45.80% ± 0.57% | 26.59% ± 0.65% |
| 3 4 8 | 44.49% ± 0.55% | 28.36% ± 0.46% |
| 3 5 6 | 45.49% ± 0.71% | 26.89% ± 0.90% |
| 3 5 7 | 48.14% ± 0.73% | 23.66% ± 0.58% |
| 3 5 8 | 46.61% ± 0.69% | 25.90% ± 0.39% |
| 3 6 7 | 44.01% ± 0.72% | 28.20% ± 0.58% |
| 3 6 8 | 42.00% ± 0.65% | 29.93% ± 0.33% |
| 3 7 8 | 45.21% ± 0.43% | 27.26% ± 0.41% |
| 4 5 6 | 41.84% ± 0.76% | 30.20% ± 0.53% |
| 4 5 7 | 44.20% ± 0.72% | 27.64% ± 0.48% |
| 4 5 8 | 42.31% ± 0.82% | 29.81% ± 0.33% |
| 4 6 7 | 40.02% ± 0.71% | 32.55% ± 0.58% |
| 4 6 8 | 37.66% ± 0.49% | 31.94% ± 0.38% |
| 4 7 8 | 40.96% ± 0.62% | 30.78% ± 0.56% |
| 5 6 7 | 42.77% ± 0.60% | 29.66% ± 0.69% |
| 5 6 8 | 40.55% ± 0.68% | 31.02% ± 0.47% |
| 5 7 8 | 43.79% ± 0.91% | 28.54% ± 0.55% |
| 6 7 8 | 38.95% ± 0.57% | 31.59% ± 0.66% |
| max (=layerX) | 50.48% ± 0.67% (2 3 5) | 31.94% ± 0.38% (4 6 8) |
| min (=layerX) | 37.66% ± 0.49% (4 6 8) | 21.71% ± 0.60% (2 3 5) |

Table 34: `UnitMem` **distinguishes between individual examples within a class.** We use 1000 samples for each experiment to compute the UnitMem score. **All**: denotes all classes, **TPC**: stands for the 3 following classes Truck, Plance, and Car classes, while **Car**: is simply the car class.

| Layer Number | All min | All max | All avg | TPC min | TPC max | TPC avg | Car min | Car max | Car avg |
|---|---|---|---|---|---|---|---|---|---|
| **Layer 1** | 0±0 | 0.845±0.014 | 0.366±0.011 | 0.007±1e-4 | 0.801±0.018 | 0.357±0.009 | 0.011±1e-4 | 0.829±0.015 | 0.360±0.010 |
| **Layer 2** | 0.006±9e-5 | 0.832±0.016 | 0.352±0.010 | 0±0 | 0.789±0.015 | 0.350±0.013 | 0.009±8e-5 | 0.810±0.011 | 0.351±0.013 |
| **Layer 3** | 0±0 | 0.841±0.017 | 0.363±0.008 | 0±0 | 0.800±0.014 | 0.355±0.010 | 0.010±9e-5 | 0.825±0.009 | 0.356±0.008 |
| **Layer 4** | 0±0 | 0.871±0.012 | 0.377±0.009 | 0.004±1e-4 | 0.833±0.019 | 0.373±0.012 | 0.015±2e-4 | 0.844±0.014 | 0.371±0.009 |
| **Layer 5** | 0.010±2e-4 | 0.859±0.016 | 0.381±0.008 | 0.016±3e-4 | 0.810±0.013 | 0.375±0.011 | 0.013±1e-4 | 0.837±0.012 | 0.380±0.010 |
| **Layer 6** | 0.020±4e-4 | 0.905±0.018 | 0.403±0.011 | 0.022±3e-4 | 0.868±0.019 | 0.381±0.008 | 0.030±5e-4 | 0.879±0.014 | 0.394±0.007 |
| **Layer 7** | 0.019±3e-4 | 0.894±0.013 | 0.398±0.009 | 0.021±3e-4 | 0.859±0.014 | 0.380±0.013 | 0.019±3e-4 | 0.861±0.017 | 0.387±0.011 |
| **Layer 8** | 0.017±2e-4 | 0.905±0.013 | 0.409±0.010 | 0.25±4e-4 | 0.863±0.017 | 0.385±0.010 | 0.024±4e-4 | 0.870±0.013 | 0.397±0.012 |

