# OpenReview forum: "Localizing Memorization in SSL Vision Encoders"
_NeurIPS.cc/2024/Conference — NeurIPS 2024 poster_

### Official Review · Reviewer_2UD6 · 2024-07-04

**Soundness:** 3
**Presentation:** 3
**Contribution:** 3
**Rating:** 7
**Confidence:** 4

**Summary:**

This work focuses on memorization in self-supervised learning. This paper seems to extent the work from [1] on SSLMem, that leverage the fact that a given training data and its corresponding handcrafted image augmentations should have a lower distance in the SSL embedding space than with a model in which this point was not in the training data. The authors extend SSLMem by using intermediate layers instead of the final representation. By doing so, the authors can localize which layers are the most memorizing training data. They show that indeed the last layer memorize the most, however we can still find significant memorization occurring also in intermediate layers. The author improve then their method by introducing a memorization metric based on individual units instead of a layer basis.

[1] Memorization in self-supervised learning improves downstream generalization, Wang et al, ICLR 2024

**Strengths:**

The paper is very well written and the authors provide extensive experiments.  The author provide results on different architecture, SSL methods, training criteria and augmentations. The results are strong.

**Weaknesses:**

- The paper can be read as just an extension of [1], so this contribution could be seen as lacking novelty. However, the empirical analysis that is provided might still interest the research community.
- I would be careful with claim such as being "first metric for localizing memorization". When we read Meehan et al., we can see that they did an ablation study across different layers, so they seem to be able to analyze which layer memorize the most. So it seems that there should have been a closer analysis and comparison with this paper results. In addition, they also provide experiments with ViT, so the author might also downplay the claim that investigating memorization in ViT is lacking.  So one weakness of this paper is the lack of empirical comparison with respect to the current literature.
- Even if the authors did a great job in trying different models, they did not analyze on a sample level basis, how the memorized examples might differ depending on random seed or different hyper-parameters.  Like for two models that produces similar LayerMem score, are they memorizing the same examples/data points or not? Because both the overall score could be similar while memorizing different examples. A similar question is, does the memorize examples change across different layers? I would have appreciated reading at least some discussion on the relationship between model memorization with respect to which examples and how much those examples are consistent between models and layers.
- Concerning the SSLMem metric, I am wondering what is the interplay between both models (training with and without specific points) initialization and hyper-parameters. Like I would suspect that depending on the data shuffling seed, the examples that might be memorized first in the early layers might differ.
- There is also a lock of discussion on the impact of normalization layer or weight decay on the experimental results. I would suspect that having a stronger weight decay might reduce memorization.

**Questions:**

- When replacing random or most/less memorized layers, how do you deal with normalization layers like batch norm since the statistics might be off?
- How much time does it take to train both models, compute augmentations and get the LayerMem and UnitMem metric?

**Limitations:**

The authors provide a limited limitation section. I would have appreciated to see as limitation that the method require training two disjoint model while having access to the original training data augmentation set. Thus, it does not seems that such method might work well on public model for which we do not know well the training details. Also the lack of a sample level analysis of the memorized example is an important limitation.

---

> ### Author Rebuttal · Authors · 2024-08-06
>
> >**W1: Novelty vs. extension of [1]**
>
> The paper targets an orthogonal research question to [1]. While [1] is concerned with the question of **quantifying** memorization and  **identifying memorized samples**, we answer the question of **where inside the SSL encoders** is the information stored.
> While doing so, we make multiple original contributions as summarized in [W3 for Reviewer Reviewer BVAM](https://openreview.net/forum?id=R46HGlIjcG&noteId=3o0QWN7snN).
>
> >**W2a: Claims on being first metric for localizing memorization in SSL**
>
> We wanted to frame our work as being the first metrics *particularly designed* for localizing memorization and the first that allow localization down to the level of individual units. To better express this framing according to the reviewer’s insightful remark, we altered the following sentences (change bolded):
> - **Introduction:** "We propose LayerMem and UnitMem, **two** practical metrics that allow to localize memorization in SSL encoders on a per-layer basis and, **for the first time, down to the granularity of individual units.**”
> - **Conclusion:** "We propose **[REMOVED]** practical metrics for localizing memorization within SSL encoders on a per-layer basis, and, **for the first time, down to the individual-unit level.**”
>
> Additionally, we added in line 107 "**The same trend was also reported for SSL [36]**" to give credit to Meehan et al. [36] for their contribution to the field (see their Appendix A7).
>
> >**W2b: Downplaying claim on memorization in ViTs**
>
> We also changed this claim in the paper (changes in bold):
> - **Introduction:** “Yet, with our methods to measure memorization, we **are able** to show that the same trend holds in vision transformers that was previously reported for language transformers, namely that memorization happens in the fully-connected layers.”
> - **Line 217:** “The **localization of** memorization **in** transformers was primarily investigated in the language domain **[REMOVED]** **and** the methods for analysis and the findings are not easily transferable. Yet, with our methods to localize memorization, we **[REMOVED]** show that the same trend holds in vision transformers that was previously reported for language transformers, namely that memorization happens in the fully-connected layers.”
>
> >**W2c: Empirical comparison to prior work**
>
> The two existing works on memorization in SSL [36,47] (SSLMem and SSL Deja Vu) consider an orthogonal research question, namely **quantifying** memorization and **identifying memorized samples**. We, in contrast, focus on **localizing** memorized content. An empirical comparison does not seem sensible.
>
> Yet, we can compare the trends reported by the ablation from Meehan et al. [36]  (their Appendix A7: Deja Vu memorization for 4 different layers of VICReg) to ours. They report that Deja Vu Memorization increases monotonically and occurs mainly in the last two layers. A similar trend for *accumulated* memorization is observed by our LayerMem, while our Delta LayerMem shows that the actual increase is not monotonic.
>
> >**W3a: Difference in memorized samples between two different encoders**
>
> This interesting research question is orthogonal to our work where we identify **where in the encoder** training data is memorized.
>
> >**W3b: Difference in memorized samples across different layers**
>
> We performed a new analysis in Figure 2 and Table 3 (attached PDF). The overlap within the 100 most memorized samples between adjacent layers is usually high but decreases the further the layers are separated. Our statistical analysis to compare the similarity of the orderings within different layers’ most memorized samples using the Kendall’s rank correlation coefficient shows that while for closer layers, we manage to reject the null hypothesis (“no correlation”) with high statistical confidence (low p value) which is not the case for further away layers.
>
>
> >**W4: Hyperparameters and variation in memorized samples.**
>
> We performed an additional experiment where we trained encoders f and g independently with a different random seed (yielding f’ and g’). We compared the overlap in most memorized samples between encoder f (from the paper) and f’.
> The results (Table 4, attached PDF) show that overlap is overall high (min. 69% in layer 2) and increases in the later layers (max. 90%, final layer).
>
>
> >**W5: Weight decay**
>
> We performed the experiment suggested by the reviewer and trained ResNet9 using SimCLR on CIFAR10 with three different levels of weight decay. Our results in Figure 1 (attached PDF) show that stronger weight decay yields lower memorization, yet also decreases linear probing accuracy.
>
>
> >**Q1: BatchNorm**
>
> We measured the cosine similarity between the weights and biases of the BatchNorm layers for two encoders (trained on CIFAR10 and STL10). The results in Table 5 (attached PDF) show a high per-layer cosine similarity (average over all layers=0.823). This suggests that the statistics are not far off, hence, no adjustment is required. We hypothesize that the similarity stems from the fact that the data distributions are similar and that we normalize both input datasets according to the ImageNet normalization parameters.
>
> >**Q2: Compute times**
>
> Following our standard setup from Table 1 and Figure 1 from the main paper, we report the following timing on an Nvidia RTX4090 (24G), 64GB RAM (for ResNet9  trained 600/200 epochs):
>
>  |Train a model|Compute UnitMem|Compute LayerMem|
> |-|-|-|
> |195min/63min|5min26sec|5min34sec|
>
> >**Limitation 1: Need of two models**
>
> This limitation only holds for LayerMem. UnitMem operates directly on the given encoder without the need for a second model. We added the limitation of LayerMem to Appendix F.
>
> >**Limitation 2: Lack of sample-level analysis**
>
> This is an orthogonal research question to this work. We do not aim to study memorization from the sample but from the encoder perspective, asking the question **where** inside the encoder training data is stored.

---

> > ### Author Response · Authors · 2024-08-09
> > **Have concerns been addressed?**
> >
> > We would like to thank the Reviewer for recognizing that our work enables localizing memorization, down to the granularity of individual units.
> > In summary, our rebuttal addressed the following:
> > 1. **Adjusting the claims**: We adjusted the claims according to the reviewer’s suggestion to recognize Meehan et al.’s contributions on analyzing ViTs and performing the ablation study on which layers hold highest accumulated Deja Vu memorization.
> > 2. **Comparing memorization between layers**: Based on the reviewer’s suggestion, we performed additional experiments to study how memorized samples differ between the layers of a single encoder and between the layers of two different encoders. We show that especially adjacent layers memorize similar data points, and that over different encoders, the overlap of most memorized samples is most similar in later layers.
> > 3. **Weights decay**: We performed also additional experiments based on the reviewer’s suggestion to show how UnitMem changes under different strength of weight decay and found that memorization decreases as weight decay increases, yet, also downstream performance drops.
> >
> > We would like to check if there are any pending concerns that should address to further strengthen the reviewer’s confidence for acceptance of our paper?

---

> ### Comment · Reviewer_2UD6 · 2024-08-12
>
> I would like to thank the authors for addressing my concerns. The rebuttal and new experiments provide interesting insights and will make the paper stronger. Therefore, I increase score.

---

### Official Review · Reviewer_GSFs · 2024-07-13

**Soundness:** 3
**Presentation:** 3
**Contribution:** 3
**Rating:** 6
**Confidence:** 3

**Summary:**

This paper introduces two novel metrics, LayerMem and UnitMem, to measure where memorization occurs within self-supervised neural networks. Through extensive experiments, this paper finds that memorization increases with layer depth, highly memorizing units are distributed throughout the encoder, atypical data points cause higher memorization, and most memorization happens in fully connected layers in ViTs. These findings suggest that localizing memorization can enhance fine-tuning and inform pruning strategies for better model performance and efficiency.

**Strengths:**

- This paper proposes the first metric to measure memorization in a self-supervised learning context.
- It presents several interesting and novel findings. For example, it's insightful that the fully connected layers in ViTs memorize the dataset more than the self-attention layers, like in NLP tasks. Also, it is interesting to see that memorization does not significantly depend on the self-supervised learning methods employed.
- Overall, the paper is well-written and logically organized.

**Weaknesses:**

- The primary limitations of this paper are the lack of in-depth analysis and practical insights. Although the paper provides a broad range of empirical findings, it could be significantly improved by exploring how and why the neural networks behave as they do. Alternatively, strongly linking to or elaborating on prior work might be a promising direction to enhance the paper. Moreover, it would be beneficial if the paper provided practical methods to improve self-supervised neural networks based on the given takeaways. Even though Section 4.2 was interesting and insightful, it is insufficient for applying these findings in a real-world setting.
- If I understand correctly, I am not fully convinced by the definitions of `LayerMem` and `UnitMem` as appropriate metrics to measure memorization. For example, according to Equation 1, LayerMem seems to depend on the neural network’s properties. If neural networks are sufficiently equivariant with respect to data augmentation, then SSL and SSLMem should be zero. However, fortunately, the equivariance of ViTs is on par with that of CNNs [1], so I believe the main takeaways can hold in this case. Also, these metrics might depend on the norms of intermediate representations.
- There is room for improvement in the writing. The paper claims that “memorization increases but not monotonically,” yet I observed that LayerMem increases monotonically and even uniformly, except in some layers, as shown in Tables 1, 14, 15, and 16. Some interesting results, e.g., Lines 247 and 387, are only included in the Appendix. The paper attempts to correlate atypical datasets and memorization, but the evidence does not strongly support the claims. It is a minor thing, but I’m not familiar with ‘SL’ as an abbreviation for supervised learning.

Overall, considering the paper as one of the first attempts to investigate memorization in a self-supervised learning setting, I don’t find major weaknesses. Despite some limitations, I lean toward acceptance since I believe such attempts should be encouraged.

[1] Gruver, Nate, et al. "The lie derivative for measuring learned equivariance." arXiv preprint arXiv:2210.02984 (2022).

**Questions:**

- In Figure 2, why does SSL UnitMem start with a non-zero value even at the first layer? I expected it would start near zero. Also, the values appear to be almost the same across all layers.
- Although high LayerMem values are observed in the later layers, this might be insufficient to conclude that these layers memorize more than the early layers. I anticipated that the effect of memorization would accumulate as the network depth increases.
- Please refer to the weaknesses section for additional comments.

**Limitations:**

Please refer to the weaknesses section. No ethical concerns found.

---

> ### Author Rebuttal · Authors · 2024-08-05
>
> >**W1a: In depth analysis and practical insights**
>
> We thank the reviewer for their suggestion. We went through the paper and identified multiple sections for adding practical insights. We summarized them in the [first reply to W1 for Reviewer BVAM](https://openreview.net/forum?id=R46HGlIjcG&noteId=3o0QWN7snN).
>
> If reviewer see some other parts in the paper where we could add more analyses (as well as intuitions and potential implications), we’d be grateful for the suggestions.
>
> >**W1b: Applying findings in real-world settings**
>
> In addition to Section 4.2, we also provide a further real-world application of our metrics by showing that UnitMem can effectively be used to inform compression schemes. In particular, it seems to be most helpful to prune the least memorizing units (vs. random units), see Table 5. On the contrary, the most memorizing units should be preserved.
>
> >**W2a: For example, according to Equation 1, LayerMem seems to depend on the neural network’s properties. If neural networks are sufficiently equivariant with respect to data augmentation, then SSL and SSLMem should be zero.**
>
> Could the reviewer further clarify what they mean by SSL (and SSLMem) should be zero? We do agree that a perfectly (over)fitted SSL-trained encoder f can have a zero representation alignment, i.e. the representations of two different augmentations of the same input sample x are identical. However, SSLMem and therefore also our LayerMem rely on an additional reference model g and report the difference in representation alignment between f and g.
>
> In case g has not seen x but still achieves perfect representation alignment means that the distribution of other data is sufficient to perfectly fit x, which also means that f does not need to memorize x. Then indeed LayerMem will be zero, correctly indicating no memorization. However, if the representation alignment for g is worse than for f, their difference will be non-zero, hence LayerMem will also be non-zero.
>
> >**W2b: Also, these metrics might depend on the norms of intermediate representations.**
>
> To address the reviewer’s concern, we performed additional measurements on the outputs of each convolution block in ResNet50 trained with SimCLR on CIFAR10. In particular, we measured the alignment loss ($\ell_2$ distance between the representations of two different augmentations) on the encoder f and g used for LayerMem. This is the signal that the metric operates on (not directly on the representations), and hence, its magnitude (norm) would influence the metric.
>
> In the Table below, we observe that while the alignment loss first increases and then drops significantly, LayerMem increases continuously, suggesting that LayerMem does not depend on the norm of the representation alignment.
> The magnitude of the norm could also be influenced by the dimensionality of the representation. Therefore, we also report output dimensionality. We observe that the dimensionality of the representations after the 1st block and the 4th block are the same. However, the memorization score is much higher for the layer after the 4th block highlighting that our LayerMem is also independent of the output dimensionality.
>
>  | Output of model block  | LayerMem | Number of Dimensions in the representations | Alignment loss difference for f: d (f (x′), f (x′′)) | Alignment loss difference for g: d (g(x′), g(x′′)) |
> |:-:|:-:|:-:|:-:|:-:|
> |1(conv1)|0.046±0.006|64\*56\*56=200704|163.25|167.56|
> |2(conv2-9)|0.103±0.014|256\*56\*56=802816|304.65|315.98|
> |3(conv3-12)|0.165±0.007|512\*28\*28=401408|187.33|221.82|
> |4(conv4-18)|0.279±0.011|1024\*14\*14=200704|98.21|129.6|
> |5(conv5-9)|0.335±0.013|2048\*7\*7=100352|61.01|84.92|
>
>
> >**W3a: There is room for improvement in the writing. The paper claims that “memorization increases but not monotonically,” yet I observed that LayerMem increases monotonically and even uniformly [...].**
>
> The quote used in the review “memorization increases but not monotonically” is not taken from our paper. Our sole statement about monotonicity in layer memorization is (in line 197) “Delta LayerMem indicates that the memorization increases in all the layers but is not monotonic.” We agree that LayerMem would indicate a monotonic increase (Table 1). As we note in the paper (line 195) “However, our Delta LayerMem presents the memorization from a more accurate perspective, where we discard the accumulated memorization from previous layers, including the residual connections.” Looking at the Delta LayerMem column in Table 1, we see that values go from 0.029->0.002->0.027…. which is not a monotonic increase. We changed the above statement to  “Delta LayerMem indicates that the memorization increases **overall with deeper layers** but it is not monotonic.”
>
> >**W3b: Some interesting results, e.g., Lines 247 and 387, are only included in the Appendix.**
>
> We appreciate the reviewer’s suggestion on which results should be additionally featured in the main body of the paper and would use the additional page in case of acceptance for including the “Verification of Layer-Based Memorization” from Appendix C.6 and the “Additional Verification of UnitMem” experiments from Appendix C.1.
>
> >**W3c: The paper attempts to correlate atypical datasets and memorization, but the evidence does not strongly support the claims.**
>
> Relating atypical examples with memorization is not a contribution from our work. This relation has been established, e.g. by [23] and [47]. We only state their findings.
>
>
> >**Overall, considering the paper as one of the first attempts to investigate memorization in a self-supervised learning setting, I don’t find major weaknesses. Despite some limitations, I lean toward acceptance since I believe such attempts should be encouraged.**
>
> We thank the reviewer for their encouraging feedback and are happy to provide further insights and clarifications that would strengthen the reviewer’s confidence for an acceptance.

---

> > ### Author Response · Authors · 2024-08-09
> > **Have concerns been addressed?**
> >
> > We would like to thank the Reviewer for recognizing our work as the first to propose metrics for localization of memorization in SSL encoders and presenting insightful as well as interesting findings.
> > In summary, our rebuttal addressed the following:
> > 1. **In depth analysis**: We extended the analysis of the paper and included the study on which samples are memorized in which layers. As requested, we added more explanations to the paper (using the additional 1 page if accepted).
> > 2. **Validating the metrics**: We provide additional experimental and conceptual insights to validate our metrics, in particular, we show that the design of LayerMem is independent of the norms in the intermediate representations.
> >
> > We would like to check: are there any pending concerns that we should address to further strengthen the reviewer’s confidence for acceptance of our paper?

---

> > > ### Comment · Reviewer_GSFs · 2024-08-11
> > > **Official Comment of Submission1482 by Reviewer GSFs**
> > >
> > > Thank you, authors, for addressing my concerns. I still believe the paper offers interesting takeaways. Therefore, I would like to maintain my original score.

---

> > > > ### Author Response · Authors · 2024-08-11
> > > > **Thank you!**
> > > >
> > > > We are happy that the Reviewer finds interesting takeaways in our paper. Thank you for maintaining the score.

---

### Official Review · Reviewer_ne6k · 2024-07-13

**Soundness:** 2
**Presentation:** 3
**Contribution:** 2
**Rating:** 5
**Confidence:** 4

**Summary:**

This paper identifies where memorization occurs in SSLs, noting that memorization increases in deeper network layers, though high-memorizing units are distributed throughout the network. For the first time, they introduce a metric to measure memorization in SSLs and provide justification for its validity. They demonstrate that memorization primarily occurs in the fully-connected layers of vision transformers.

**Strengths:**

* This paper proposes a metric to define memorization in self-supervised models and identifies the locations where the most memorization occurs.

**Weaknesses:**

* It is not clear why the defined metric is a good measure of memorization, especially for comparing memorization across different layers. The magnitude of the distance between representations of different augmentations is highly dependent on the activation distribution of the layers and the distance metric used. While the normalization proposed in equations 2 and 4 attempts to address this, I am still unsure if it is sufficient.

* To justify the validity of the LayerMem memorization metric, they argue that effective memorization in SSLs leads to strong downstream performance. They also point out that replacing the layers with the highest memorization scores, as determined by the metric, results in a more performance drop, indicating the metric's reliability. However, I believe this justification alone is insufficient. Are there other pieces of evidence supporting the validity of this memorization metric?

**Questions:**

* When replacing the two layers of models trained on CIFAR-10 and STL-10, if the replaced layer is at the beginning of the network, don't the activations propagate through the entire network? I don't think I fully understood that experiment.

* Is L2 distance used as the distance metric? If so, what is the justification for using it, and do the results change if a different metric is used?

**Limitations:**

Yes

---

> ### Author Rebuttal · Authors · 2024-08-05
>
> We thank the reviewer for their insightful comments.
> >**W1: It is not clear why the defined metric is a good measure of memorization, especially for comparing memorization across different layers. The magnitude of the distance between representations of different augmentations is highly dependent on the activation distribution of the layers and the distance metric used. While the normalization proposed in equations 2 and 4 attempts to address this, I am still unsure if it is sufficient.**
>
> To avoid that our memorization measure overfits to a particular set of augmentations whose activations might indeed vary throughout layers, we measure our LayerMem as an expectation over different pairs of diverse augmentations. Following SSLMem, in our experiments, we calculate LayerMem as an average over 10 different randomly chosen pairs of augmentations from the full augmentation set used during training.
>
> The only pitfall that could arise would be that layers with higher output dimensionality get a higher LayerMem (simply because there is more mass in the activation). To verify that this is not the case, we report for ResNet50 trained with SimCLR on CIFAR10 the LayerMem and output dimensionality for the last layers of each block. In the Table below, we see that the dimensionality of the representations after the 1st block and the 4th block are the same. However, the memorization score is much higher for the layer after the 4th block highlighting that our LayerMem is independent of the output dimensionality.
> | Output of model block  | LayerMem | Number of Dimensions in the representations | Alignment loss difference for f: d (f (x′), f (x′′)) | Alignment loss difference for g: d (g(x′), g(x′′)) |
> |:-:|:-:|:-:|:-:|:-:|
> | 1 (conv1) | 0.046 ± 0.006 | 64\*56\*56= 200704 | 163.25 | 167.56 |
> | 2 (conv2-9) | 0.103 ± 0.014 | 256\*56\*56 = 802816 | 304.65 | 315.98 |
> | 3 (conv3-12) | 0.165 ± 0.007 | 512\*28\*28 = 401408 | 187.33 | 221.82 |
> | 4 (conv4-18) | 0.279 ± 0.011 | 1024\*14\*14 =200704 | 98.21 | 129.6 |
> |5 (conv5-9) | 0.335 ± 0.013  | 2048\*7\*7 =100352 | 61.01 | 84.92 |
>
> >**W2: To justify the validity of the LayerMem memorization metric, they argue that effective memorization in SSLs leads to strong downstream performance. They also point out that replacing the layers with the highest memorization scores, as determined by the metric, results in a more performance drop, indicating the metric's reliability. However, I believe this justification alone is insufficient. Are there other pieces of evidence supporting the validity of this memorization metric?**
>
> The finding that effective memorization leads to strong downstream performance has been proven for supervised learning by [23] and for SSL by [47].  We build on these prior findings and  show that LayerMem can help us to select layers for fine-tuning to achieve better downstream performance. Does the reviewer have any other suggestions on how we could provide additional evidence? We are happy to implement them.
>
> >**Q1: Layer Replacement**
>
> We agree that swapping affects subsequent layers. If we replace layer N, the input to layer N+1 and **all subsequent layers** is altered. Following this logic of a cascading effect, replacing the earliest layers should result in the highest impact, simply because most layers are affected by the change. However, this is not what we experimentally observe (e.g., in Table 22). Instead we see that swapping out the most memorizing layer(s) causes the largest change in the downstream performance.
>
> What is the interesting insight from the results is that the replacement of different layers has a different impact on the test accuracy of STL10. In particular, we observe that replacing most memorized CIFAR10 encoder layers with their STL10 equivalent boosts the STL10 performance significantly more than replacing, for example, the least memorized or random layers. This indicates that the layers with the highest memorization have the highest impact on the downstream performance.
>
> >**Q2: Distance Metric**
>
> Following SSLMem, we used the $\ell_2$ metric for the main paper. To address the reviewer’s questions, we performed an additional experiment reporting LayerMem with different distance metrics ($\ell_1$ distance, cosine similarity, angular distance). Our results in the table below (see also Table 1 in the attached PDF) highlight that 1) the memorization scores are very similar, independent of the choice of the distance metric, and 2) the most memorizing layers according to LayerMem and Delta Layermem are the same over all metrics. This suggests that our findings are independent of the choice of distance metric.
>
> |  | L2 (from paper) |  | L1 |  | Cosine similarity |  | Angular distance |  |
> |---|:---:|:---:|:---:|:---:|:---:|:---:|:---:|:---:|
> | Layer | LayerMem | Delta Layermem | LayerMem | Delta Layermem | LayerMem | Delta Layermem | LayerMem | Delta Layermem |
> | 1 | 0.091 | - | 0.099 | - | 0.104 |  | 0.096 | - |
> | 2 | 0.123 | 0.032 | 0.128 | 0.029 | 0.134 | 0.030 | 0.128 | 0.032 |
> | 3 | 0.154 | 0.031 | 0.159 | 0.031 | 0.163 | 0.029 | 0.160 | 0.032 |
> | 4 | 0.183 | 0.029 | 0.187 | 0.028 | 0.190 | 0.027 | 0.191 | 0.031 |
> | Res2 | 0.185 | 0.002 | 0.192 | 0.005 | 0.193 | 0.003 | 0.193 | 0.002 |
> | 5 | 0.212 | 0.027 | 0.221 | 0.029 | 0.220 | 0.027 | 0.222 | 0.029 |
> | 6 | 0.246 | **0.034** | 0.256 | 0.035 | 0.256 | **0.036** | 0.259 | **0.037** |
> | 7 | 0.276 | 0.030 | 0.289 | 0.033 | 0.288 | 0.032 | 0.293 | 0.034 |
> | 8 | 0.308 | 0.032 | 0.325 | **0.036** | 0.321 | 0.033 | 0.328 | 0.035 |
> | Res6 | **0.311** | 0.003 |**0.329** | 0.004 | **0.323** | 0.002 | **0.329** | 0.001 |

---

> > ### Author Response · Authors · 2024-08-09
> > **Have concerns been addressed?**
> >
> > We would like to thank the Reviewer for recognizing our work to proposes the first metrics for localization of memorization in SSL.
> > In summary, our rebuttal addressed the following:
> > 1. **Validity of the measure**: We present additional conceptual and experimental results validating that our metrics localize memorization.
> > 2. **Clarifications**: We provided clarification on our experimental insights.
> > 3. **Distance metric**: We performed additional experiments highlighting that our LayerMem metric is independent of the underlying distance metric.
> >
> > We would like to check: are there any other pending concerns that we should address?

---

> > > ### Comment · Reviewer_ne6k · 2024-08-11
> > >
> > > Thank you for addressing my concerns. I have increased my score.

---

> > > > ### Author Response · Authors · 2024-08-11
> > > > **Thank you!**
> > > >
> > > > We thank the Reviewer for the response and for increasing the score. We are glad that our answers addressed the concerns. Since the rating is still on the borderline, we are more than happy to provide any additional insights if needed.

---

### Official Review · Reviewer_BVAM · 2024-07-29

**Soundness:** 3
**Presentation:** 3
**Contribution:** 3
**Rating:** 7
**Confidence:** 4

**Summary:**

This paper investigates memorization in self-supervised learning encoders. It introduces two metrics, LayerMem and UnitMem, to locate memorization on a layer-wise and unit-wise basis. These metrics provide insights into the distribution of memorization within neural networks. The study reveals that memorization occurs throughout the layers of SSL encoders, not just in the final layers, and that, in vision transformers, memorization primarily takes place in fully connected layers. Furthermore, the authors propose practical applications for their metrics, such as enhancing the efficiency of fine-tuning and pruning through memorization localization.

**Strengths:**

- The empirical evaluations follow a logical flow, utilizing state-of-the-art architectures and datasets, resulting in informative results. Overall the empirical results are quite extensive and thorough.
- The proposed metrics are computationally efficient and practically convenient, requiring only a forward pass and no labels. Additionally, the UnitMem metric offers an intuitive and insightful definition.
- The paper is overall very well-written and the topic/motivation is interesting.

**Weaknesses:**

- There are some missing explanations on some of the observations. In addition to presenting empirical results, the authors should include explanations and insightful remarks to clarify their intuition and the potential implications of these results. This is done to some extent already, but would improve the paper if it is done in all the sections.
- The presentation of the results could be improved. Tables full of numbers with tiny differences after the decimal point make it hard to read and navigate over. Instead of using tables that explain themselves, the paper relies on text explanations.
- The proposed metrics, while inspired by previously known metrics, lack complete novelty. The LayerMem metric is primarily based on the existing SSLMem metric, with the only difference being a summation operation. This incremental modification does not constitute significant innovation.

**Questions:**

- I am puzzled on why the memorization pattern remains the same between different datasets. For example in Figure 1  the histogram of UnitMem remains unchanged regardless of variations in the dataset or augmentations. Why does this happen? Do the authors have some suggested explanations for this?
- Besides using these metrics in fine tuning are there other practical benefits to these proposed metrics?

**Limitations:**

The authors discuss their metrics limitations to some extent.

---

> ### Author Rebuttal · Authors · 2024-08-05
>
> >**W1: Adding insights and explanations**
>
> We thank the reviewer for their suggestion. We went through the paper and identified the following sections for adding explanations and insightful remarks within the additional page that could be added to the paper in case of acceptance:
> - **Section 4, definition of LayerMem:** We will add further insights on how the use of the reference encoder g makes LayerMem independent of the internal representation dimensionality of the encoder. Therefore, we point to our experiments showing that LayerMem is independent of the internal representation dimensionality, since we always set two encoders’ internal layers with same dimensionality in relation.
> - **Section 4.1, line 195**: We will elaborate more why metrics, like LayerMem that measure memorization in the forward pass measure an accumulated effect of memorization. This is happening since the output of the previous layer influences the current one. This motivates the study of other metrics, such as Delta LayerMem, which consider layers more independently and can lead to more fine-grained insights on memorization.
> - **Section 4.1, line 219**: We will relate the observation that “fully-connected layers memorize more than attention layers” to the number of parameters in the layers. The fully connected layers have more storage capacity, which could, potentially have an impact on privacy-preserving architectural choices in neural networks.
> - **Section 5.1, line 314**: We will explain more on the memorization patterns between different datasets and how they relate to the distribution of atypical examples in the dataset.
> - **Section 5.1, line 327**: We will add insights on how our finding–that highest memorizing units memorize the highest memorized data points from prior work–suggests that there is a particular set of neurons responsible for memorization, and that the memorized data is not distributed over all neurons within the network.
> - Following the suggestion by Reviewer 2UD6, we also added additional insights into the consistency of data memorized in different layers (see response to the reviewer and attached PDF).
> If reviewer see some other parts in the paper where we could add more explanations (as well as intuitions and potential implications), we’d be grateful for the suggestions.
>
> >**W2: Presentation of results**
>
> We appreciate the reviewer’s suggestion. We wanted the tables to convey the full picture of results. We hope that the added insights (W1) will provide guidance to the reader on how to interpret the trends. Additionally, we performed additional experiments during the rebuttal (see general response, attached PDF, and responses to the other reviewers) and we tried to include more Figures (1 and 2) that would appeal to the readers with visual preference.
>
> **W3: Novelty**
>
> While our LayerMem builds heavily on SSLMem, it is only a part of our innovations. In summary, the paper makes the following original contributions:
> 1. We introduce UnitMem as the first practical method to localize memorization in SSL encoders down to  the unit level (Section 5).
> 2. We derive Delta LayerMem from LayerMem (which is indeed based on SSLMem), inspired by our insight that LayerMem does measure accumulated memorization rather than individual layers’ memorization.
> 3. We perform thorough studies on localizing memorization in layers and units and leverage these insights to 1) inform the design on more effective fine-tuning approaches (Table 4) and 2) identify possible future pruning strategies to reduce both memorization and model complexity (Table 5).
>
> > **Q1a: Memorization pattern over different datasets**
>
> While the distributions of UnitMem scores (Figure 1a) look similar, the main difference is in the number of highly memorizing units (those that exhibit high values of UnitMem): the SVHN dataset, which is visually less complex than the CIFAR10 or STL10 dataset, has the lowest number of highly memorizing units. Additionally, all encoders considered in this work are trained with self-supervised learning objectives, yielding a similar memorization pattern.
>
> > **Q1b: Memorization pattern over different augmentations**
>
> Note that the different augmentation sets presented in the submission are only used at the inference time only **during the calculation of UnitMem** (see Equation 5), not during training. To study how different augmentations during training affect UnitMem, we performed additional experiments using **different augmentation strengths during training**.
> The results, reported in Table 2 in the attached PDF highlight that stronger training augmentations lead to higher UnitMem. We hypothesize that this effect is caused by the model having to incorporate more information on the training samples to make up for the scarser input information (through the stronger augmentation).
>
> >**Q2: Practical Benefits**
>
> Our results suggest that UnitMem can also inform memorization-based compressing schemes for neural networks. As shown in Table 5, pruning the least memorized units preserves better downstream performance than just removing random units (while removing memorizing units harms downstream performance most).
> We further hypothesize that UnitMem could yield a signal to strengthen white-box membership attacks against SSL encoders and that the knowledge of which neuron memorizes which data point from training could be used as a form of model watermark.

---

> > ### Author Response · Authors · 2024-08-09
> > **Have concerns been addressed?**
> >
> > We would like to thank the Reviewer for recognizing our work as interesting, practical, and well executed. The paper has definitely improved as a result of the feedback.
> > In summary, our rebuttal addressed the following:
> > 1. **Additional Insights**: We improved the writing by adding additional insights, observations, explanations, and insightful remarks.
> > 2. **Augmentations**: We performed additional requested experiments that show that stronger training augmentations lead to higher UnitMem.
> > 3. **Practical Application**: We provided a list of practical applications for our work, where apart from improved fine-tuning, it could also be used for the memorization-based compressing schemes, strengthening the membership inference attacks, and watermarking.
> >
> > We would like to check if there are any pending concerns that would further strengthen the reviewer’s confidence for acceptance of our paper?

---

> > ### Comment · Reviewer_BVAM · 2024-08-09
> > **Higher memorization with stronger data augmentation**
> >
> > I would like to thank the authors for their response and running additional experiments.
> >
> > In particular for the data augmentation experiment, I am puzzled by the results. The results show that the stronger the data augmentation during training, the higher memorization happens in a layer level.
> >  Why is this the case? Shouldn't it be the opposite? Why the units/layers experience more memorization if during training they have seen more data samples? Data augmentation would in principle decrease memorization.
> >
> >
> > (btw, I think there is a typo in your response? I only see the results in the attached PDF for LayerMem but in the response it is written the results are for UnitMem?)

---

> > > ### Author Response · Authors · 2024-08-09
> > > **Augmentation Strength and Memorization**
> > >
> > > We thank the reviewer for engaging in the discussion with us and are happy to answer the questions.
> > >
> > > >**Augmentation Strength and Memorization.**
> > >
> > > First, we would like to apologize for the confusion regarding the reported metric. Indeed, in the attached PDF, we considered LayerMem (while we wrote UnitMem in the rebuttal).
> > >
> > > Based on the reviewer’s comment, we ran additional experiments to verify our results. This time, for verification with an independent metric, we *actually* computed the average per-layer **UnitMem** for encoders trained with different strengths of augmentations. The results occurred to be the opposite to what we reported on LayerMem in the previous comment. Unfortunately, we made a mistake in the previous experiment with LayerMem due to the heat of the rebuttal and we are terribly sorry for that. When computing the LayerMem score, we did not change the path to the reference model $g$ in the code (and always took the $g$ trained with normal augmentation strength), even when evaluating $f$ trained with weak or strong augmentations.
> > >
> > > As a consequence, when evaluating memorization with the strong augmentations, our $g$--trained with normal augmentations--had an unusually high alignment loss, resulting in the reported memorization being too high. In contrast, when evaluating for weak augmentations, our $g$--again trained with normal augmentations--had unusually small alignment loss, resulting in the reported memorization being too low.
> > >
> > > We corrected the experimental setup and re-ran the experiment. The new results for LayerMem and UnitMem are aligned and demonstrate that memorization is actually smaller for stronger augmentations:
> > >
> > > ### **Results for LayerMem:**
> > >
> > > | augmentations: | weak |  | normal (from paper) |  | strong |  |
> > > |:---:|:---:|:---:|:---:|:---:|:---:|:---:|
> > > | Layer | LayerMem | Delta Layermem | LayerMem | Delta Layermem | LayerMem | Delta Layermem |
> > > | 1 | 0.092 | - | 0.091 | - | 0.089 | - |
> > > | 2 | 0.123 | 0.031 | 0.123 | 0.032 | 0.120 | 0.031 |
> > > | 3 | 0.154 | 0.031 | 0.154 | 0.031 | 0.150 | 0.030 |
> > > | 4 | 0.184 | 0.030 | 0.183 | 0.029 | 0.178 | 0.028 |
> > > | Res2 | 0.187 | 0.003 | 0.185 | 0.002 | 0.181 | 0.003 |
> > > | 5 | 0.215 | 0.028 | 0.212 | 0.027 | 0.208 | 0.027 |
> > > | 6 | 0.249 | 0.034 | 0.246 | 0.034 | 0.241 | 0.033 |
> > > | 7 | 0.280 | 0.031 | 0.276 | 0.030 | 0.269 | 0.028 |
> > > | 8 | 0.313 | 0.033 | 0.308 | 0.032 | 0.300 | 0.031 |
> > > | Res6 | 0.315 | 0.002 | 0.311 | 0.003 | 0.302 | 0.002 |
> > >
> > > ### **Results for UnitMem:**
> > >
> > > | augmentations: | weak | normal (from paper) | strong |
> > > |:---:|:---:|:---:|:---:|
> > > | Layer | Avg. UnitMem | Avg. UnitMem | Avg. UnitMem |
> > > | 1 | 0.362 | 0.360 | 0.355 |
> > > | 2 | 0.357 | 0.354 | 0.352 |
> > > | 3 | 0.361 | 0.357 | 0.351 |
> > > | 4 | 0.365 | 0.362 | 0.355 |
> > > | Res2 | - | - | - |
> > > | 5 | 0.370 | 0.366 | 0.360 |
> > > | 6 | 0.375 | 0.374 | 0.364 |
> > > | 7 | 0.381 | 0.379 | 0.369 |
> > > | 8 | 0.387 | 0.384 | 0.375 |
> > > | Res6 | - | - | - |
> > >
> > > Please note that since the residual connections do not contain individual units, it is impossible to compute the UnitMem score for them.
> > >
> > > The new results for *augmentations* now align with the effects observed when applying **regularization** through weight decay. As shown in Figure 1 in the attached rebuttal PDF, we observe that with stronger weight decay regularization, memorization also decreases.
> > >
> > > Once again, we sincerely apologize for the confusion and hope that the new results fully address the concerns.

---

### Author Rebuttal · Authors · 2024-08-05

We would like to thank all the reviewers for their insightful comments and questions. We are happy that the reviewers recognize our work: “presents several interesting and novel findings” (Reviewer GSFs). We are also glad the reviewers appreciate our new metrics to be “computationally efficient and practically convenient” (Reviewer BVAM)”, and our paper to provide thorough and extensive experiments (Reviewer BVAM, Reviewer 2UD6) which are presented in a well-written paper (Reviewer BVAM, Reviewer GSFs).

**Highlights of the rebuttal:**

During the rebuttal, we performed additional experiments and analyses that we present in the individual answers to the reviewers and in summary in the attached PDF.
1. Based on the suggestion by Reviewer ne6k, we show that while in the paper, we follow SSLMem and use the $\ell_2$ distance to compute LayerMem, our LayerMem is insensitive to the choice of the distance metric, and all additional metrics ($\ell_1$ Distance, Cosine Similarity, Angular Distance) yield similar results and trends (see Table 1 in the attached PDF).
2. While previously in the paper, we had only experimented with varying the augmentations used to compute UnitMem *during inference*, based on the suggestion by Reviewer BVAM, we varied the augmentations used *during training* and show that stronger augmentations yield higher memorization in individual units according to UnitMem  (see Table 2 in PDF).
3. Based on the suggestion by Reviewer 2UD6, we experimented additionally with varying the weight decay during training and find that with stronger weight decay, we observe less memorization at the costs of decreased downstream performance (see Figure 1 in PDF).
4. Finally, to provide further insights, in accordance with the suggestion by Reviewer 2UD6, we analyzed the variability and consistency between the samples memorized by different layers in the model. Our results (see Figure 2 and Table 3 in PDF) show the interesting trends that the most memorized samples vary between layers (more the further the layers are away from one another), and tend to be more similar between adjacent layers, in particular the last layers of the model.

---

### Decision · Program_Chairs · 2024-09-25

**Decision:**

Accept (poster)

**Comment:**

This paper investigates memorization in self-supervised learning encoders. It introduces two metrics, LayerMem and UnitMem, to locate memorization on a layer-wise and unit-wise basis. These metrics provide insights into the distribution of memorization within neural networks. The study reveals that memorization occurs throughout the layers of SSL encoders, not just in the final layers, and that, in vision transformers, memorization primarily takes place in fully connected layers. Furthermore, the authors propose practical applications for their metrics, such as enhancing the efficiency of fine-tuning and pruning through memorization localization.

Please update the paper with the clarifications and discussions that took place during rebuttal period.